# Continual Invariant Risk Minimization

## Abstract

Empirical risk minimization can lead to poor generalization behaviour on unseen environments if the learned model does not capture invariant feature representations. Invariant risk minimization (IRM) is a recent proposal for discovering environment-invariant representations. It was introduced by Arjovsky et al. (2019) and extended by Ahuja et al. (2020). The assumption of IRM is that all environments are available to the learning system at the same time. With this work, we generalize the concept of IRM to scenarios where environments are observed sequentially. We show that existing approaches, including those designed for continual learning, fail to identify the invariant features and models across sequentially presented environments. We extend IRM under a variational Bayesian and bilevel framework, creating a general approach to continual invariant risk minimization. We also describe a strategy to solve the optimization problems using a variant of the alternating direction method of multiplier (ADMM). We show empirically using multiple datasets and with multiple sequential environments that the proposed methods outperforms or is competitive with prior approaches.

## 1 Introduction

Empirical risk minimization (ERM) is the predominant principle for designing machine learning models. In numerous application domains, however, the test data distribution can differ from the training data distribution. For instance, at test time, the same task might be observed in a different environment. Neural networks trained by minimizing ERM objectives over the training distribution tend to generalize poorly in these situations. Improving generalization of learning systems has become a major research topic in recent years, with many different threads of research including, but not limited to, robust optimization (e.g., Hoffman et al. (2018)) and domain adaptation (e.g., Johansson et al. (2019)). Both of these research directions, however, have their own intrinsic limitations (Ahuja et al. (2020)). Recently, there have been proposals of approaches that learn environment-invariant representations. The motivating idea is that the behavior of a model being invariant across environments makes it more likely that the model has captured a causal relationship between features and prediction targets. This in turn should lead to a better generalization behavior. Invariant risk minimization (IRM, Arjovsky et al. (2019)), which pioneered this idea, introduces a new optimization loss function to identify non-spurious causal feature-target interactions. Invariant risk minimization games (IRMG, Ahuja et al. (2020)) expands on IRM from a game-theoretic perspective.

The assumption of IRM and its extensions, however, is that all environments are available to the learning system at the same time, which is unrealistic in numerous applications. A learning agent experiences environments often sequentially and not concurrently. For instance, in a federated learning scenario with patient medical records, each hospital's (environment) data might be used to train a shared machine learning model which receives the data from these environments in a sequential manner. The model might then be applied to data from an additional hospital (environment) that was not available at training time. Unfortunately, both IRM and IRMG are incompatible with such a continual learning setup in which the learner receives training data from environments presented in a sequential manner. As already noted by Javed et al. (2020), "IRM Arjovsky et al. (2019) requires sampling data from multiple environments simultaneously for computing a regularization term pertinent to its learning objective, where different environments are defined by intervening on one or more variables of the world." The same applies to IRMG (Ahuja et al. (2020))

To address the problem of learning environment-invariant ML models in sequential environments, we make the following contributions:

- We expand both IRM and IRMG under a Bayesian variational framework and develop novel objectives (for the discovery of invariant models) in two scenarios: (1) the standard multi-environment scenario where the learner receives training data from all environments at the same time; and (2) the scenario where data from each environment arrives in a sequential manner.
- We demonstrate that the resulting bilevel problem objectives have an alternative formulation, which allows us to compute a solution efficiently using the alternating direction method of multipliers (ADMM).
- We compare our method to ERM, IRM, IRMG, and various continual learning methods (EWC, GEM, MER, VCL) on a diverse set of tasks, demonstrating comparable or superior performance in most situations.

## 2   BACKGROUND: OFFLINE INVARIANT RISK MINIMIZATION

We consider a multi-environment setting where, given a set of training environments $E = \{e_1, e_2, \cdots, e_m\}$, the goal is to find parameters $\theta$ that generalize well to unseen (test) environments. Each environment $e$ has an associated training data set $D_e$ and a corresponding risk $R^e$

$$R^e(w \circ \phi) \doteq E_{(x,y) \sim D_e} \ell_e((w \circ \phi)(x), y), \tag{1}$$

where $f_\theta = w \circ \phi$ is the composition of a feature extraction function $\phi$ and a classifier (or regression function) $w$. Empirical Risk Minimization (ERM) minimizes the average loss across all training examples, regardless of environment:

$$R_{\mathrm{ERM}}(\theta) \doteq E_{(x,y) \sim \cup_{e \in E} D_e} \ell(f_\theta(x), y). \tag{2}$$

ERM has strong theoretical foundations in the case of iid data (Vapnik (1992)) but can fail dramatically when test environments differ significantly from training environments. To remove spurious features from the model, Invariant Risk Minimization (IRM, Arjovsky et al. (2019)) instead aims to capture invariant representations $\phi$ such that the optimal classifier $w$ given $\phi$ is the same across all training environments. This leads to the following multiple bi-level optimization problem

$$\min_{\phi \in H_\phi, w \in H_w} \sum_{e \in E} R^e(w \circ \phi) \quad \text{s.t. } w \in \arg\min_{w_e \in H_w} R^e(w_e \circ \phi), \forall e \in E, \tag{3}$$

where $H_\phi, H_w$ are the hypothesis sets for, respectively, feature extractors and classifiers. Unfortunately, solving the IRM bi-level programming problem directly is difficult since solving the outer problem requires solving multiple dependent minimization problems jointly. We can, however, relax IRM to IRMv1 by fixing a scalar classifier and learning a representation $\phi$ such that the classifier is "approximately locally optimal" (Arjovsky et al. (2019))

$$\min_{\phi \in H_\phi} \sum_{e \in E} R^e(\phi) + \lambda ||\nabla_{w|w=1.0} R^e(w\phi)||^2, \forall e \in E, \tag{4}$$

where $w$ is a scalar evaluated in 1 and $\lambda$ controls the strength of the penalty term on gradients on $w$. Alternatively, the recently proposed Invariant Risk Minimization Games (IRMG) (Ahuja et al. (2020)) proposes to learn an ensemble of classifiers with each environment controlling one component of the ensemble. Intuitively, the environments play a game where each environment's action is to decide its contribution to the ensemble aiming to minimize its risk. Specifically, IRMG optimizes the following objective:

$$\min_{\phi \in H_\phi} \sum_{e \in E} R^e(\bar{w} \circ \phi) \quad \text{s.t. } w_e = \arg\min_{w \in H_w} R^e\left(\frac{1}{|E|}(w + w_{-e}) \circ \phi\right), \forall e \in E, \tag{5}$$

where $\bar{w} = \frac{1}{|E|} \sum_{e \in E} w_e$ is the average and $w_{-e} = \sum_{e' \in E, e' \neq e} w_{e'}$ the complement classifier.

## 3   CONTINUAL IRM BY APPROXIMATE BAYESIAN INFERENCE

Both IRM and IRMG assume the availability of training data from all environments at the same time, which is impractical and unrealistic in numerous applications. A natural approach would be

to combine principles from IRM and continual learning. Experience replay, that is, memorizing examples of past environments and reusing them later, could be possible in some scenarios but it is often difficult to estimate a-priori the extend of replay necessary to achieve satisfactory generalization capabilities. Here, we propose to adopt a probabilistic approach, exploiting the propagation of the model distribution over environments using Bayes' rule. We integrate both IRM and IRMG with stochastic models, introducing their variational counterparts that admit a continual extension. In addition, our approach is justified by the property of the Kullback–Leibler (KL) divergence that promotes invariant distributions when used in sequential learning (as shown in Theorem 3).

## 3.1 VARIATIONAL CONTINUAL LEARNING

Following prior work in continual learning (Nguyen et al. (2018)), let $D_t$ be the training data from the $t$-th environment $e^t$, let $D_1^t$ be the cumulative data up to the $t$-th environment, and let $\theta$ be the parameters of the feature extractor. When each environment is given in a sequential manner, we can use Bayes' rule and we have (all proofs are provided in the supplementary material)

$$p(\theta|D_1^t) \propto p(\theta|D_1^{t-1})p(D_t|\theta), \tag{6}$$

that is, once we have the posterior distribution $p(\theta|D_1^{t-1})$ at time $t-1$, we can obtain, by applying Bayes rule, the posterior $p(\theta|D_1^t)$ at time $t$ up to a normalization constant. This is achieved by multiplying the previous posterior with the current data likelihood $p(D_t|\theta)$. The posterior distribution is in general not tractable and we use an approximation. With the variational approximation, $p(\theta|D_1^t) \approx q_t(\theta)$, it is thus possible to propagate the variational distribution from one environment to the next. From Corollary 14 (in the supplementary material) we can write the continual variational Bayesian inference objective as

$$q_t(\theta) = \arg\min_{q(\theta)} \mathbb{E}_{(x,y)\sim D_t} \mathbb{E}_{\theta\sim q(\theta)} \{\ell(y, f_\theta(x))\} + D_{\mathrm{KL}}(q(\theta)||q_{t-1}(\theta)), \tag{7}$$

from the variational distribution at step $q_{t-1}(\theta)$, with $f_\theta = w \circ \phi$, a function with parameters $\theta$.

## 3.2 EQUIVALENT FORMULATION OF IRM AS A BILEVEL OPTIMIZATION PROBLEM (BIRM)

In order to extend the IRM principle of Equation 3 using the principle of approximate Bayesian inference, by applying Lemma 5 (in supplementary material), we first introduce the following new equivalent definition of IRM (equation 3).

**Definition 1** (Bilevel IRM (BIRM)). *Let $H_\phi$ be a set of feature extractors and let $H_w$ be the set of possible classifiers. An **invariant predictor** $w \circ \phi$ on a set of environments $E$ is said to satisfy the Invariant Risk Minimization (IRM) property, if it is the solution to the following bi-level Invariant Risk Minimization (BIRM) problem*

$$\min_{\phi\in H_\phi, w\in H_w} \sum_{e\in E} R^e(w \circ \phi) \qquad (8a) \qquad s.t. \quad \nabla_w R^e(w \circ \phi) = 0, \forall e \in E. \qquad (8b)$$

This formulation results from substituting the minimization conditions in the constraint set of the original IRM formulation with the Karush–Kuhn–Tucker (KKT) optimality conditions. This new formulation allows us to introduce efficient solution methods and simplifies the conditions of IRM. It also justifies the IRMv1 model; indeed, when the classifier is a scalar value and the equality constraint is included in the optimization cost function, we obtain Equation 4. To solve the BIRM problem, we propose to use the Alternating Direction Method of Multipliers (ADMM) (Boyd et al. (2011)). ADMM is an alternate optimization procedure that improves convergence and exploits the decomposability of the objective function and constraints. Details of the BIRM-ADMM algorithm are presented in the supplementary material.

## 3.3 BILEVEL VARIATIONAL IRM

At this point, we cannot yet directly extend the IRM principle using variational inference. That is because if we observe all environments at the same time, the prior of the single environment is data independent. Therefore, we substitute $q_{t-1}(\theta)$ from Equation 7 with priors $p_\phi(\theta)$ and $p_w(\omega)$, where $\theta$ and $\omega$ are now the parameters of the two functions $\phi$ and $w$. We also substitute $q_t(\theta)$ with the variational distributions $q_\phi(\theta)$ and $q_w(\omega)$.

**Definition 2** (Bilevel Variational IRM (BVIRM)). *Let $P_\phi$ be a family of distributions over feature extractors, and let $P_w$ be a family of distributions over classifiers. A **variational invariant predictor** on a set of environments $E$ is said to satisfy Bilevel Variational Invariant Risk Minimization (BVIRM) if it is the solution to the following problem:*

$$\min_{\substack{q_\phi \in P_\phi \\ q_w \in P_w}} \sum_{e \in E} Q_\phi^e(q_w, q_\phi) \quad (9\text{a}) \qquad s.t. \ \nabla_{q_w} Q_w^e(q_w, q_\phi) = 0, \forall e \in E, \quad (9\text{b})$$

$$with \qquad Q_\phi^e(q_w, q_\phi) = \mathbb{E}_{\substack{w \sim q_w \\ \phi \sim q_\phi}} R^e(w \circ \phi) + \beta D_{\mathrm{KL}}(q_\phi || p_\phi) + \beta D_{\mathrm{KL}}(q_w || p_w), \quad (10\text{a})$$

$$and \qquad Q_w^e(q_w, q_\phi) = \mathbb{E}_{\substack{w \sim q_w \\ \phi \sim q_\phi}} R^e(w \circ \phi) + \beta D_{\mathrm{KL}}(q_w || p_w), \quad (10\text{b})$$

*and where $p_\phi$ and $p_w$ are the priors of the two distributions. $\beta$ is a hyper-parameter balancing the ERM and closeness to the prior.*

Definition 2 extends Definition 1 with the objective of Eq.7, where the parameters $\phi$ and $w$ are substituted by their distributions $q_\phi$ and $q_w$. The gradient of the cost in the inner problem is taken with respect to the distribution $q_w$. When we parameterize $q_\phi$ with $\theta$ and $q_w$ with $\omega$, the gradient is evaluated with respect to these parameters[1], since the condition implies that the solution is locally optimal. If $Q(p, q)$ is convex in the first argument, then the solution is globally optimal. This definition extends the IRM principle to the case where we use approximate Bayes inference, shaping the variational distributions $q_w$ and $q_\phi$, to be, in expectation, invariant and optimal across environments.

## 3.4 THE BVIRM ADMM ALGORITHM

As noted for the BIRM definition, the solution of the variational BVIRM formulation can be obtained by using ADMM (Boyd et al. (2011)). While in general there are no convergence results of ADMM methods for this problem, for local minima, under proper conditions [2], the stochastic version of ADMM converges with rate $O(1/\sqrt{t})$ for convex functions and $O(\log t/t)$ for strongly convex functions (Ouyang et al. (2013)). We are now in the position to write the BVIRM-ADMM formulation of the BVIRM problem. ADMM is defined by the update Eq.11, where we denote with the apexes $^-$ and $^+$ the value of any variable before and after the update. Moreover, we abbreviate as follows $Q(\omega, \theta) = Q(q(\omega), q(\theta))$.

$$\omega_e^+ = \arg\min_{\omega_e} L_\rho(\omega_e, u_e^-, \omega^-, v_e^-), \forall e \in E, \quad (11\text{a})$$

$$\omega^+ = 1/|E| \sum_e (\omega_e + u_e) \quad (11\text{b})$$

$$u_e^+ = u_e^- + (\omega_e^+ - \omega^+) \quad (11\text{c})$$

$$v_e^+ = v_e^- + \nabla_{q(\omega)} Q_w^e(\omega_e^+, \theta) \quad (11\text{d})$$

with

$$L_\rho(w_e, u_e, w, v_e) = Q_\phi^e(\omega_e, \theta) + \frac{\rho_0}{2} \|\omega_e - \omega + u_e\|^2 + \frac{\rho_1}{2} \|\nabla_{q(\omega)} Q_w^e(\omega_e \circ \phi) + v_e\|^2. \quad (12)$$

Here, $\phi$ is fixed and $\theta$ is updated in an external loop or given (e.g. the identity function). In the experiment we use stochastic Gradient Descent (SGD) to update both the model parameters $w_e$ and the feature extractor parameters $\phi$. The result follows by applying Lemma 11 in the supplementary material and substituting $x_i \leftarrow \begin{pmatrix} w_e \\ \phi \end{pmatrix}$, $f_i(x_i) \leftarrow Q_\phi^e(\omega_e, \theta)$ and $g_i(x_i) \leftarrow \nabla_{q(\omega)} Q_w^e(\omega_e^+, \theta)$. We provide a pseudo-code implementation leveraging Equation 11 as Algorithm 1. One of the advantages of the ADMM formulation of BVIRM of Eq.11, is that it can be computed in parallel, where only Eq.11b requires synchronization among environments, while the other steps can be computed independently.

---

[1] Implementation detail using the mean field parameterization and reparametrization trick is provided in the Supplementary Material [2] These conditions are specific bounds on the magnitude and variance of the (sub-)gradients of the stochastic function (Ouyang et al. (2013)). We used ELU $\in C^\infty$ in the experiments.

**Algorithm 1:** $w, \phi \leftarrow$ BVIRM-ADMM$(E, R^e)$ ADMM version of the Bilevel Variational IRM Algorithm

**Result:** $w \circ \phi$ : feature extraction and classifier for the environment $E$
// Randomly initialize the
variables
1 $\omega, \omega_e, u_e, v_e, \theta \leftarrow$ Init() ;
// Outer loop (on $\theta$) and
Inner loop (on $\omega$)
2 **while** *not converged* **do**
3     // Update $\phi$ using SGD
    $\theta = \text{SGD}_\theta(\sum_{e \in E} Q_\phi^e(q_w, q_\phi))$ ;
4     **for** $k = 1, \dots, K$ **do**
5         **for** $e \in E$ **do**
6             $\omega_e = \text{SGD}_{\omega_e} L_\rho(\omega_e, u_e, \omega, v_e)$ ;
7             $\omega = 1/|E| \sum_e (\omega_e + u_e)$ ;
8             $u_e = u_e + (\omega_e - \omega)$ ;
9             $v_e = v_e + \nabla_\omega Q^e(\omega_e, \theta)$ ;
10         **end**
11     **end**
12 **end**

**Algorithm 2:** $w, \phi \leftarrow$ C-BVIRM-ADMM$(E, R^e)$ ADMM version of the Bilevel Variational IRM Algorithm

**Result:** $w_\omega \circ \phi - \theta$ : feature extraction and classifier for the environment $E$
// Randomly initialize the
variables
1 $\omega, \omega_e, u_e, v_e, \theta \leftarrow$ Init() ;
2 $\bar{\omega} = 0$ ;
3 **for** $e \in E$ **do**
4     **for** $k = 1, \dots, K$ **do**
5         $\theta = \text{SGD}_\theta(Q_\phi^e(q_w, q_\phi))$ ;
6         **while** *not converged* **do**
            // Update $\omega$ using SGD
            and ADMM
7             $\omega_e = \text{SGD}_{\omega_e} L_\rho(\omega_e, u_e, \omega, v_e)$ ;
8             $\omega = 1/2(\omega_e + u_e + \bar{\omega})$ ;
9             $u_e = u_e + (\omega_e - \omega)$ ;
10             $v_e = v_e + \nabla_\omega Q^e(\omega_e, \theta)$ ;
11         **end**
12     **end**
13     $\bar{\omega} = \omega_e$ ;
14 **end**

### 3.4.1 THE CONTINUAL BVIRM ADMM ALGORITHM

In presence of sequential environments, the priors for the new environment are given by the previous environment's distributions $q_\phi^-$ and $q_w^-$, this is obtained by comparing the BVIRM definition in Eqs. (9) with the continual Bayesian learning Equation (7). In Equation 10 we thus now have $Q_\phi^e(q_w, q_\phi) = \mathbb{E}_{w \sim q_w, \phi \sim q_\phi} R^e(w \circ \phi) + \beta D_{\text{KL}}(q_\phi || q_\phi^-) + \beta D_{\text{KL}}(q_w || q_w^-)$ and $Q_w^e(q_w, q_\phi) = \mathbb{E}_{w \sim q_w, \phi \sim q_\phi} R^e(w \circ \phi) + \beta D_{\text{KL}}(q_w || q_w^-)$ Algorithm 2 presents an example implementation of ADMM[3] applied to the continual BVIRM formulation.

### 3.5 INFORMATION-THEORETIC INTERPRETATION OF C-BVIRM

The KL divergence provides an additional motivation for the methods we propose. Indeed, for causal discovery Peters et al. (2015) suggests a discovery mechanism for causal variables as the intersection of the invariant conditional distributions across environments subject to interventions. The KL divergence is asymmetric and only components present in the first argument distribution are evaluated. This implies that by using the KL divergence we can compute the intersection of the distributions, even when these are observed sequentially. This can be made more explicit by the property of the information projection Cover (1999)

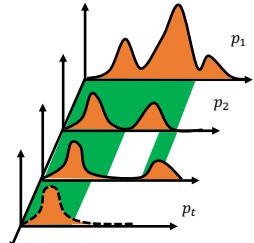

**Theorem 3** (Information Projection). *If $P$ and $Q$ are two families of distributions with partially overlapping support, $\emptyset \subset \text{supp}(P) \bigcap \text{supp}(Q)$, and $q \in Q$, then*

$$p^* = \arg\min_{p \in P} D_{\text{KL}}(p || q)$$

Figure 1: Sequential projection of distributions $p_1, \dots p_t$, where $p_{i+1} = \arg\min_{p \in P_{i+1}} D_{\text{KL}}(p || p_i)$

*has support in the intersection for the support of $P$ and $q$, or* $\text{supp}(p^*) \subseteq \text{supp}(P) \bigcap \text{supp}(q)$.

---

[3] In Algorithm 1 the ADMM update equation is implemented from line 6 to line 9, while in Algorithm 2, from line 7 to line 10.

Figure 2: The two color models (on the left `b01`, on the right `b11`) for the train (upper row) and test (lower row) of the MNIST (left) and FashionMNIST (right) datasets.

Therefore, if we have a sequence of sets of distributions of models from intervention environments and we compute the projection in sequence, the final projected distribution has support on the intersection of all previous distribution families, or $\mathrm{supp}(P_t) = \bigcap_{i=1}^{t} \mathrm{supp}(P_i)$ (see Figure 1) , since at each step $p_{i+1} = \arg\min_{p \in P_{i+1}} D_{\mathrm{KL}}(p||p_i)$ .

## 4 RELATED WORK

**Generalization** Domain adaptation (Ben-David et al., 2007; Johansson et al., 2019) aims to learn invariant features or components $\phi(x)$ that have similar $P(\phi(x))$ on different (but related) domains by explicitly minimizing a distribution discrepancy measure, such as the Maximum Mean Discrepancy (MMD) (Gretton et al., 2012) or the Correlation Alignment (CORAL) (Sun & Saenko, 2016). The above condition, however, is not sufficient to guarantee successful generalization to unseen domains, even when the class-conditional distributions of all covariates changes between source and target domains (Gong et al., 2016; Zhao et al., 2019). Robust optimization (Hoffman et al., 2018; Lee & Raginsky, 2018), on the other hand, minimizes the worst performance over a set of possible environments $E$, that is, $\max_{e \in E} R^e(\theta)$. This approach usually poses strong constraint on the closeness between training and test distributions (Bagnell, 2005) which is often violated in practical settings (Arjovsky et al., 2019; Ahuja et al., 2020).

Incorporating the machinery of causality into learning models is a recent trend for improving generalization. (Bengio et al., 2019) argued that causal models can adapt to sparse distributional changes quickly and proposed a meta-learning objective that optimizes for fast adaptation. IRM, on the other hand, presents an optimization-based formulation to find non-spurious actual causal factors to target $y$. Extensions of IRM include IRMG and the Risk Extrapolation (REx) (Krueger et al., 2020). Our work's motivation is similar to that of online causal learning (Javed et al., 2020), which models the expected value of target $y$ given each feature as a Markov decision process (MDP) and identifies the spurious feature $x_i$ if $\mathbb{E}[y|x_i]$ is not consistent to temporally distant parts of the MDP. The learning is implemented with a gating model and behaves as a feature selection mechanism and, therefore, can be seen as learning the support of the invariant model. The proposed solution, however, is only applicable to binary features and assumes that the aspect of the spurious variables is known (e.g. the color). It also requires careful hyper-parameter tuning. In the cases where data is not divided into environments, Environment Inference for Invariant Learning (EIIL) classification method (Creager et al. (2020)) aims at splitting the samples into environments. This method proves to be effective also when the environment label is present.

**Continual Learning** Kirkpatrick et al. (2017); De Lange et al. (2019) addresses the problem of learning one classifier that performs well across multiple tasks given in a sequential manner. The focus is on the avoidance of catastrophic forgetting. With our work, we shift the focus of continual learning to the study of a single task that is observed in different environments.

## 5 EXPERIMENTAL EVALUATION

### 5.1 DATASETS AND EXPERIMENT SETUP

**Colored MNIST** Figure 2 (left) shows a sample of train (upper) and test (lower) samples. In each training environment, the task is to classify whether the digit is, respectively, even or odd. As in prior work, we add noise to the preliminary label by randomly flipping it with a probability

Table 1: Mean accuracy ($N = 5$) on train and test environments when training on 2 consecutive environments on MNIST and the b01 color correlation.

| | C-BVIRM | C-VIRMG | C-VIRMv1 | ERM | EWC | GEM | IRMG | IRMv1 | MER | VCL | VCLC |
|---|---|---|---|---|---|---|---|---|---|---|---|
| train | 71.3 | 69.1 | 51.4 | 86.4 | 87.4 | 87.4 | 86.3 | 85.3 | 87.3 | **89.3** | **89.3** |
| | (4.2) | (2.8) | (3.4) | (1.2) | (2.7) | (2.7) | (1.2) | (0.8) | (1.7) | (0.6) | (0.6) |
| test | 29.6 | 27.9 | **46.0** | 12.7 | 15.7 | 15.7 | 12.8 | 9.9 | 14.8 | 24.9 | 24.9 |
| | (3.3) | (8.5) | (2.1) | (2.7) | (4.2) | (4.2) | (2.6) | (0.2) | (3.8) | (1.9) | (1.9) |

of $0.25$. The color of the image is defined by the variable $z$, which is the noisy label flipped with probability $p_c \in [0.1, 0.2]$. The color of the digit is green if $z$ is even and red if $z$ is odd. Each train environment contains $30,000$ images of size $28 \times 28$ pixels, while the test environment contains $10,000$ images where the probability $p_c = 0.9$. The color of the digit (b01) or the background (b11) is thus generated from the label but depends on the environment. Figure 3 depicts the causal graph (the hammer indicating the effect of the intervention) of the environment. The variable "Color" is inverted when moving from the training to test environment.

**Colored FashionMNIST, KMNIST, and EMNIST**  Figure 2 (right) shows the Fashion-MNIST dataset, where the variable $z$ defines the background color. Again, we add noise to the preliminary label ($y = 0$ for "t-shirt", "pullover", "coat", "shirt", "bag" and $y = 1$ for "trouser", "dress", "sandal", "sneaker", "ankle boots") by flipping it with 25 percent probability to construct the final label. Besides, we also consider Kuzushiji-MNIST dataset Clanuwat et al. (2018)[4] and the EMNIST Letters dataset Cohen et al. (2017)[5]. The former includes 10 symbols of Hiragana, whereas the latter contains 26 letters in the modern English alphabet. For EMNIST, there are $62,400$ training samples per environment and $20,300$ test samples. We set $y = 0$ for letters 'a', 'c', 'e', 'g', 'i', 'k', 'm', 'o', 'q', 's', 'u', 'v', 'y' and $y = 1$ for remaining ones.

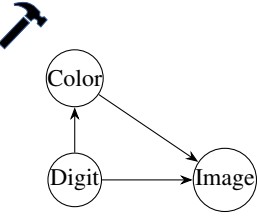

Figure 3: Causal relationships of colored MNIST.

**Reference Methods.** We compare with a set of popular reference methods in order to show the advantage of the variational Bayesian framework in learning invariant models in the sequential environment setup. For completeness, we also evaluate the performances of four reference continual learning methods. These include Elastic Weight Consolidation (EWC, Kirkpatrick et al. (2017)), Gradient Episodic Memory (GEM, (Lopez-Paz & Ranzato, 2017))[6], Meta-Experience Replay (MER, Riemer et al. (2018))[7], and Variational Continual Learning (VCL, Swaroop et al. (2019); Nguyen et al. (2018))[8]. **ERM** is the classical empirical risk minimization method; we always use the cross-entropy loss. **IRMv1** enforces the gradient of the model with respect to a scalar to be zero. **IRMG** models the problem as a game among environments, where each environment learns a separate model. **EWC** imposes a regularization cost on the parameters that are relevant to the previous task, where the relevance is measured by Fisher Information (FI); **GEM** uses episodic memory and computes the updates such that accuracy on previous tasks is not reduced, using gradients stored from previous tasks; **MER** uses an efficient replay memory and employs the meta-learning gradient update to obtain a smooth adaptation among tasks; **VCL** and Variational Continual Learining with coreset **VCLC** apply variational inference to continual learning. **C-VIRMv1** and **C-VIRMG** refer to, respectively, our proposed variational extensions of **IRMv1** and **IRGM** in sequential environments. **C-BVIRM** is the implementation with ADMM.

All hyper-paramter optimization strategies and simulation configurations are discussed in detail in the supplementary material.

---

[4] https://github.com/rois-codh/kmnist          [5] https://www.nist.gov/itl/products-and-services/emnist-dataset
[6] https://github.com/facebookresearch/GradientEpisodicMemory          [7] https://github.com/mattriemer/mer
[8] https://github.com/nvcuong/variational-continual-learning

Table 2: Mean accuracy (over 5 runs) and standard deviation at test time for (n) $2, 6, 10$ environments, (d) across datasets, and (c) for the two color correlations (b01,b11).

| | | C-BVIRM | C-VIRMG | C-VIRMv1 | ERM | EWC | GEM | IRMG | IRMv1 | MER | VCL | VCLC |
|---|---|---|---|---|---|---|---|---|---|---|---|---|
| n | 2 | 29.6 | 27.9 | **46.0** | 12.7 | 15.7 | 15.7 | 12.8 | 9.9 | 14.8 | 24.9 | 24.9 |
| | | (3.3) | (8.5) | (2.1) | (2.7) | (4.2) | (4.2) | (2.6) | (0.2) | (3.8) | (1.9) | (1.9) |
| | 6 | 28.8 | 27.5 | **47.1** | 11.1 | 15.6 | 15.6 | 12.2 | 9.7 | 15.3 | 15.4 | 15.4 |
| | | (4.1) | (2.6) | (3.0) | (2.9) | (5.7) | (5.7) | (2.8) | (0.3) | (4.4) | (1.1) | (1.1) |
| | 10 | 21.8 | 25.2 | **31.0** | 10.2 | 17.7 | 17.7 | 12.4 | 10.2 | 15.5 | 10.8 | 10.8 |
| | | (2.4) | (4.5) | (7.1) | (0.2) | (5.5) | (5.5) | (2.1) | (0.2) | (4.1) | (0.2) | (0.2) |
| d | MNIST | 29.6 | 27.1 | **46.8** | 12.7 | 15.7 | 15.7 | 12.8 | 9.9 | 14.8 | 24.9 | 24.9 |
| | | (3.3) | (7.6) | (2.6) | (2.7) | (4.2) | (4.2) | (2.6) | (0.2) | (3.8) | (1.9) | (1.9) |
| | Fa-MNIST | 36.3 | 26.7 | **48.2** | 10.7 | 15.4 | 15.3 | 10.8 | 9.9 | 13.2 | 24.9 | 24.9 |
| | | (4.3) | (8.7) | (3.6) | (1.5) | (5.1) | (5.4) | (1.4) | (0.2) | (2.8) | (2.0) | (2.0) |
| | KMNIST | 32.8 | 24.2 | **46.5** | 12.0 | 14.0 | 14.0 | 12.1 | 9.9 | 15.6 | 24.9 | 24.9 |
| | | (4.6) | (6.0) | (1.9) | (2.1) | (3.5) | (3.5) | (2.4) | (0.2) | (4.1) | (2.0) | (2.0) |
| | EMNIST | 32.1 | 25.0 | **45.9** | 10.8 | 15.3 | 14.8 | 10.8 | 10.0 | 12.6 | 24.9 | 24.9 |
| | | (4.6) | (7.5) | (2.5) | (1.0) | (3.6) | (3.7) | (1.2) | (0.2) | (2.3) | (2.0) | (2.0) |
| c | b01 | 29.6 | 27.1 | **46.8** | 14.9 | 18.8 | 18.8 | 14.6 | 9.8 | 18.0 | 24.9 | 24.9 |
| | | (3.3) | (7.6) | (2.6) | (0.8) | (1.3) | (1.3) | (0.8) | (0.1) | (1.0) | (2.0) | (2.0) |
| | b11 | 38.8 | 23.9 | **43.3** | 9.9 | 12.5 | 12.5 | 9.8 | 9.9 | 11.5 | 24.9 | 24.9 |
| | | (4.2) | (6.9) | (4.5) | (0.2) | (3.6) | (3.6) | (0.1) | (0.2) | (2.0) | (2.0) | (2.0) |

## 5.2 RESULTS

Table 1 lists the training and test accuracy on the MNIST dataset with the color correction b01 (see Figure 2 left). Since we introduced label noise by randomly flipping 25 percent of the given labels, a hypothetical optimal classifier would be able to achieve an accuracy of $75\%$ in both training and test environments. ERM, IRMv1, and IRMG perform poorly in the setup where environments are given sequentially. Similarly, reference continual learning methods also fail to learn invariant representation in the new environment. As these models are learning to mainly use spurious features for the classification problems at hand, here: the colors of the digits (red~odd; green~even), they perform poorly (much worse than a random baseline) when the spurious feature properties are inverted (green~odd; red~even). In contrast, our variational extensions to both IRM and IRMG achieve a classification accuracy higher than $45\%$ on the test data. This implies that our model is not relying exclusively on spurious correlations present in the color of digits. By comparing the performance between C-VIRMv1 and C-BVIRM, we conclude that (1) our proposed bilevel invariant risk minimization framework (i.e., the BIRM in Definition 1) is an effective alternative to the original formulation Arjovsky et al. (2019); and (2) ADMM is effective in solving the BIRM optimization problem and has the potential to improve the generalization performance. In addition, one can observe that the KL divergence term in VCL and our framework significantly improves the test accuracy with respect to the baseline counterparts. This result further justifies our motivation of using a variational Bayesian framework for the problem of continual invariant risk minimization. Table 2 lists the accuracy on the test environment for: (n) (upper rows) an increasing number of sequential environments (d) (central rows) different datasets, and (c) (lower rows) the two given color correlation schemes. We can observe that there is a general trend in the results. IRMG and IRM, with an accuracy of less than $10\%$, are not able to learn invariant models. Similarily, the continual learning reference methods (MER, EWC, MER, VCL, VCLC) also fail with a test accuracy of under $25\%$. The proposed methods on the other hand provide mechanism to learn more robust features and classification models. The higher variance of the accuracy is caused by the stochastic nature of the variational Bayesian formulation.

## 5.3 ENVIRONMENT INFERENCE FOR CONTINUAL INVARIANT LEARNING

In practical applications, the environmental labels are usually unavailable, which means that it is difficult or impossible to manually partitioning the training set into "domains" or "environments".

Table 3: Mean accuracy (over 10 runs) on train and test environments when training off-line on 2 environments on Colored-MNIST, with the EIIL. ($pc_1 = 0.2, pc_2 = 0.1, 50'000$ samples)

|  | IRMv1 | | C-VIRMv1 | |
|---|---|---|---|---|
|  | Train | Test | Train | Test |
| No EIIL | 70.73 (1.16) | **67.48** (1.96) | **70.99** (0.90) | 66.60 (2.66) |
| EIIL | 73.78 (0.61) | 67.96 (3.01) | **75.29** (0.53) | **68.40** (1.11) |

Table 4: Mean accuracy (over 5 runs) on train and test environments when training on 1 environment on Colored-MNIST, with and without EIIL. ($pc_1 = 0.1$)

|  | Without Environment Inference | | | | With Environment Inference (EIIL) | | | |
|---|---|---|---|---|---|---|---|---|
|  | IRMv1 | | C-VIRMv1 | | IRMv1 | | C-VIRMv1 | |
| $N_s$ | Train | Test | Train | Test | Train | Test | Train | Test |
| 1'000 | 93.7 (0.7) | 13.5 (1.7) | 94.1 (1.1) | 13.7 (1.5) | 95.5 (0.3) | 12.7 (2.0) | 96.0 (0.4) | 16.5 (6.0) |
| 2'000 | 91.5 (0.4) | 12.7 (0.9) | 91.1 (0.7) | 11.9 (0.9) | 92.6 (0.4) | 27.8 (2.8) | 93.3 (0.7) | 29.3 (3.4) |
| 5'000 | 90.2 (0.4) | 10.5 (1.1) | 90.1 (0.4) | 10.6 (0.7) | 91.6 (0.4) | 29.6 (4.8) | 91.6 (0.9) | 30.6 (3.2) |
| 10'000 | 89.9 (0.3) | 10.1 (0.5) | 90.0 (0.2) | 10.1 (0.1) | 85.3 (1.0) | 42.9 (3.9) | 83.7 (1.2) | 50.4 (2.3) |
| 20'000 | 90.0 (0.2) | 10.1 (0.2) | 90.1 (0.2) | 10.1 (0.0) | 77.2 (1.2) | 57.4 (2.2) | 77.9 (1.1) | 57.6 (2.0) |
| 50'000 | 90.1 (0.1) | 9.7 (0.4) | 90.0 (0.1) | 10.0 (0.4) | 73.9 (0.5) | 67.2 (1.2) | 74.0 (0.5) | 67.3 (1.0) |

In order to generalize our continual invariant learning models to an environment-agnostic setting, we leverage the recently proposed Environment Inference for Invariant Learning (EIIL) by Creager et al. (2020) to automatically infer environment partitions from observational training data, and integrate EIIL into our continual invariant learning models.

We take our proposed C-VIRMv1 as an example. According to Table 3, it is easy to observe that inferring environments directly from observational data (using EIIL) has the potential to improve (continual) invariant learning relative to using the hand-crafted environments. Moreover, C-VIRMv1 with EIIL improves both training and test accuracy, compared with IRMv1 with EIIL. In fact, this environment partition strategy also enables invariant learning with only one environmental data. Table 4 further suggests that the generalization accuracy improves for both IRMv1 and C-VIRMv1 as the number of training samples increases. Again, we observed that, when combined with EIIL, C-VIRMv1 always outperforms IRMv1.

## 6 CONCLUSIONS

We aim to broaden the applicability of IRM to settings where environments are observed sequentially. We show that reference approaches fail in this scenario. We introduce a variational Bayesian approach for the estimation of the invariant models and a solution based on ADMM. We evaluate the proposed approach with reference models, including those from continual learning, and show a significant improvement in generalization capabilities.

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

## A    SUPPLEMENTARY MATERIAL

### A.1    VARIATIONAL INVARIANT RISK MINIMIZATION GAMES

We now consider the IRMG objective and extend it with the variational Bayesian inference. If we observe all environment at the same time, the prior of the single environment is data independent. From Equation 7, we thus substitute $q_{t-1}(\theta)$ with a priors $p_\phi(\theta)$ and $q_w(\omega)$, where $\theta$ and $\omega$ are now the parameters of the two functions $\phi$ and $w$. While we substitute $q_t(\theta)$, with the variational distributions $q_\phi(\theta)$ and $q_w(\omega)$. The outer problem is now

$$\min_{q_\phi} \quad \mathbb{E}_{\phi \sim q(\phi)} R^e(\bar{w} \circ \phi) + \beta D_{\mathrm{KL}}(q_\phi || p_\phi) \tag{13a}$$

$$\text{s.t.} \quad q_{w_e} = \arg\min_{q_{w_e}} \mathbb{E}_{w \sim q_{w_e}} R^e(\frac{1}{|E|}(w + w_{-e}) \circ \phi) + \beta D_{\mathrm{KL}}(q_{w_e} || p_w) \forall e \in E^{\mathrm{tr}} \tag{13b}$$

where $\bar{w} = \frac{1}{|E|} \sum_{e \in E^{\mathrm{tr}}} w_e$, $w_e \sim q_{w_e}(w)$ is the average classifier and $w_{-e} = \sum_{e' \in E^{\mathrm{tr}}, e' \neq e} w_{e'}, w_{e'} \sim q_{w_{e'}}(w)$ is the complement classifier. In the reformulation of the IRMG model, we weight the distance of the varional distribution to the prior with $\beta$. We notice how the difference of the variational formulation of IRMG differs on the presence of the mean on the distribution of the function over the variational distributions and the KL term.

We can now finally extend IRMG when the environments are observed sequentially. Combining the definition of IRMG Eqs. (5) with the continual bayesian learning Equation (7), we obtain the variational objective of IRMG in sequential environment case.

$$\min_{q_\phi} \quad \mathbb{E}_{\phi \sim q(\phi)} \{\ell(y, \bar{w} \circ \phi)\} + \beta D_{\mathrm{KL}}(q_\phi || q_\phi^{t-1}) \tag{14a}$$

$$\text{s.t.} \quad \bar{w} = \frac{1}{2}(w + w_{t-1}), w \sim q_w(w), w_{t-1} \sim q_w^{t-1}(w) \tag{14b}$$

$$q_w = \arg\min_{q_w} \mathbb{E}_{w \sim q_{w_e}, \phi \sim q_\phi} \{\ell(y, \frac{1}{2}(w + w_{t-1}) \circ \phi\} + \beta D_{\mathrm{KL}}(q_{w_e} || q_w^{t-1}) \tag{14c}$$

We can similarly extend the definition of IRMv1 when all environments are seen at the same time and sequentially.

### A.2    MEAN FIELD PARAMETRIZATION AND REPARAMETRIZATION TRICK

When we want to implement Equation 11 and Equation 12 and the different variation, we use the mean field approximation and the reparametrization trick Kingma & Welling (2013). In this case the density function of our model is parameterized by $\theta$ and $\omega$ and constraints becomes $\nabla_{q(\omega)} Q_w^e(\omega_e^+, \theta) = 0 \to \nabla_\omega Q_w^e(\omega, \theta) = 0$. If we then parametrize $\mu(\omega_\mu)$ and $\sigma(\omega_\sigma)$ the mean and standard deviation and model the distribution as $q_\omega(w) = \mu(\omega_\mu) + \epsilon\sigma(\omega_\sigma)$, with $\epsilon \sim N(0,1)$ We now want to compute the gradient (in the following we ignore the dependence on the $\phi$ and its parameters)

$$\nabla_\omega Q(\omega) = \nabla_\omega \mathbb{E}_{w \sim q(\omega)} R(w \circ \phi) + \beta \nabla_\omega D_{\mathrm{KL}}(q(\omega) || p)$$

The second term is

$$\nabla_\omega D_{\mathrm{KL}}(q || p) = \nabla_\mu D_{\mathrm{KL}}(q || p) \nabla_\omega \mu + \nabla_\sigma D_{\mathrm{KL}}(q || p) \nabla_\omega \sigma$$

with

$$\nabla_\omega \mu = 1, \nabla_\omega \sigma = \frac{1}{\epsilon}$$

$$\nabla_\mu D_{\mathrm{KL}}(q || p) = -\sigma_p^{-1}(\mu_p - \mu_q)$$

$$\nabla_\sigma D_{\mathrm{KL}}(q || p) = -\mathrm{diag}(\sigma_q)^{-1} + \mathrm{diag}(\sigma_p)^{-1}$$

where we assume $\sigma_p, \sigma_q$ to be diagonal, in this way the previous equation can be evaluated element-wise and where the $D_{\mathrm{KL}}(q || p)$ is defined as

$$D_{\mathrm{KL}}(q || p) = \ln \frac{|\Sigma_p|}{|\Sigma_q|} - n + \mathrm{tr}\{\Sigma_p^{-1} \Sigma_q\} + (\mu_p - \mu_q)^T \Sigma_p^{-1}(\mu_p - \mu_q)$$

The first term is evaluated by Monte Carlo sampling

$$\nabla_\omega \mathbb{E}_{w \sim q(\omega)} R(w) \approx \nabla_\omega \frac{1}{N} \sum_{i=1}^N R(w_i)$$

with

$$w_i = \mu(\omega) + \epsilon_i \odot \sigma(\omega)$$

and $w_i \sim N(0,1)$. Also in this case

$$\nabla_\omega \frac{1}{N} \sum_{i=1}^N R(w_i) = \nabla_\mu \frac{1}{N} \sum_{i=1}^N R(w_i) \nabla_\omega \mu + \nabla_\sigma \frac{1}{N} \sum_{i=1}^N R(w_i) \nabla_\omega \sigma$$

### A.3 THE BIRM-ADMM ALGORITHM

We observe that to solve BIRM we can use Lemma 11 and write the following algorithm

$$w_e^+ = \arg\min_{w_e} L_\rho(w_e, u_e^-, w^-, v_e^-), \forall e \in E \tag{15a}$$

$$w^+ = 1/|E| \sum_i (w_e + u_e) \tag{15b}$$

$$u_e^+ = u_e^- + (w_e^+ - w^+) \tag{15c}$$

$$v_e^+ = v_e^- + \nabla_w R^e(w_e^+ \circ \phi) \tag{15d}$$

where

$$L_\rho(w_e, u_e, w, v_e) = R^e(w_e \circ \phi) + \frac{\rho_0}{2}\|w_e - w + u_e\|^2 + \frac{\rho_1}{2}\|\nabla_w R^e(w_e \circ \phi) + v_e\|^2 \tag{16}$$

We denote $.^+, .^-$ the values of the variable after and before the update. In order to implement the method we use the SGD to update the model $w_e$ and in a outer loop updating for $\phi$.

---

**Algorithm 3:** $w, \phi \leftarrow$ BIRM-ADMM$(E, R^e)$ ADMM version of the Bilevel IRM Algorithm

**Result:** $w \circ \phi$ : feature extraction and classifier for the environment $E$
```
   // Randomly initialize the variables
 1 w, w_e, u_e, v_e, φ ← Init() ;
   // Outer (on φ) and Inner loop (on w)
 2 while not converged do
      // Update φ using stochastic gradient descent(SGD)
 3    φ = SGD_φ(∑_e R^e(w ∘ φ)) ;
 4    for k = 1, ..., K do
 5       for e ∈ E do
 6          w_e = SGD_{w_e} L_ρ(w_e, u_e, w, v_e) ;
 7          w = 1/|E| ∑_e(w_e + u_e) ;
 8          u_e = u_e + (w_e − w) ;
 9          v_e = v_e + ∇_w R^e(w_e ∘ φ) ;
10       end
11    end
12 end
```

---

### A.4 VARIATIONAL INVARIANT RISK MINIMIZATION

**Definition 4** (VIRM). *Give a set of distribution over the mapping $P_\phi$ and a distribution over the set of classifier $P_w$, a **variational invariant predictor** on a set of environments $E$ is said to satisfy the*

*Variational Invariant Risk Minimization (VIRM) if it is the solution of the following problem*

$$\min_{\substack{q_\phi \in P_\phi \\ q_w \in P_w}} \quad \sum_{e \in E} Q_\phi^e(q_w, q_\phi) \tag{17a}$$

$$\text{s.t.} \quad q_w \in \arg\min_{q_w^e \in P_w} Q_w^e(q_w, q_\phi), \forall e \in E \tag{17b}$$

$$\text{where} \quad Q_\phi^e(q_w, q_\phi) = \mathbb{E}_{\substack{w \sim q_w \\ \phi \sim q_\phi}} R^e(w \circ \phi) + \beta D_{\mathrm{KL}}(q_\phi || p_\phi) + \beta D_{\mathrm{KL}}(q_w || p_w), \tag{17c}$$

$$Q_w^e(q_w, q_\phi) = \mathbb{E}_{\substack{w \sim q_w \\ \phi \sim q_\phi}} R^e(w \circ \phi) + \beta D_{\mathrm{KL}}(q_w || p_w) \tag{17d}$$

*and $p_\phi, p_w$ are the priors of the two distributions.*

## A.5    BILEVEL ALTERNATIVE FORMULATION

We state here a general result on solving Bilevel Optimization Problems

**Lemma 5** (Bilevel Reformulation)**.**

$$\min_x \quad F(x, y) | G(x, y(x)) \leq 0 \tag{18a}$$

$$\text{s.t.} \quad y(x) \in \arg\min_y f(x, y) | g(x, y) \leq 0 \tag{18b}$$

*then we can solve the equivalent problem*

$$\min_{x,y,u} \quad F(x, y) | G(x, y(x)) \leq 0, \tag{19a}$$

$$\nabla_y L(x, y, u) = 0, \tag{19b}$$

$$u \geq 0, \tag{19c}$$

$$g(x, y) \leq 0, \tag{19d}$$

$$u^T g(x, y) = 0 \tag{19e}$$

$$L(x, y, u) \quad = \quad f(x, y) + u^T g(x, y) \tag{19f}$$

*Proof of Lemma 5 .* Lemma 5 follows by applying the Karush-Kuhn-Tucker conditions (Chapter 5 Boyd et al. (2004)) to Eq.18, where the Lagrangian function is $L(x, y, u) = f(x, y) + u^T g(x, y)$. □

**Lemma 6** (Equivalence of Definition 1)**.** *Definition 1 is equivalent to Eq. 3, the Invariant Risk Minimization.*

*Proof of Lemma 6 .* The result follows by apply Lemma 5 to Eq.3. □

**Lemma 7** (Definition 2)**.** *Definition 2 is the extension of Eq. 8, the Bilevel Invariant Risk Minimization, when the function is described by the distributions of their variable $\phi$ and $w$.*

*Proof of Lemma 7 .* The result follows by inspecting Eq. 8. The equation requires the minimisation of the aggregated loss function, which is now, from Eq.7:

$$Q_\phi^e(q_w, q_\phi) = \mathbb{E}_{\substack{w \sim q_w \\ \phi \sim q_\phi}} R^e(w \circ \phi) + \beta D_{\mathrm{KL}}(q_\phi || p_\phi) + \beta D_{\mathrm{KL}}(q_w || p_w), \tag{20}$$

where we have separated the two contributions in $\phi$ and $w$, and used genetic prior distributions $p_\phi$ and $p_w$. This is by the additive property of KL divergence:

$$D_{\mathrm{KL}}(q_\phi q_w || p_\phi p_w) = D_{\mathrm{KL}}(q_\phi || p_\phi) + D_{\mathrm{KL}}(q_w || p_w), \tag{21}$$

since we model the two distributions independently, i.e. $q_{\phi,w} = q_\phi q_w$ and $q_{\phi,w} = p_\phi p_w$. Since the classifiers' losses shall be minimal for all environments, this condition is substituted by requiring the gradient with respect to $q_w$ to be zero, $\forall e$. The gradient w.r.t. $q_w$ of the second term of Eq.20 is zero. □

### A.6 THEOREM 3 AND IRM CONNECTION

#### A.6.1 SEQUENTIAL INFORMATION PROJECTION

In Theorem.3, we show that the Information Projection (IP) shrinks the support of the output distribution.

**Lemma 8.** *If we have a sequence of families of distributions $P_i$. Let $p_1 \in P_1$ and*

$$p_{i+1} = \arg \min_{p \in P_{i+1}} D_{\mathrm{KL}}(p, p_i)$$

*then*

$$\mathrm{supp}\, p_i \subseteq \bigcap_{j \leq i} \mathrm{supp}(P_j)$$

*Proof of Lemma 8.* We have that $\forall i, \mathrm{supp}\, p_i \subseteq \mathrm{supp}(P_i) \bigcap \mathrm{supp}(P_{i-1})$, where the first condition follows from $p_i \in P_i$ in the minimization and the second from Theorem 3. The results follows by iterating the property. $\square$

#### A.6.2 IRM AND INFORMATION PROJECTION

We show now two ways to state the connection of the IRM principle and the sequential IP. Let $q^-$ be the distribution of the previous environment and $R(q)$ the loss function of the current environment, where $q$ denotes the distribution of the network parameters. Let

$$q^* = \arg \min_q R(q)$$

be the optimal distribution for the current environment. We can then consider the Taylor expansion of the parameters distribution around the optimal distribution as

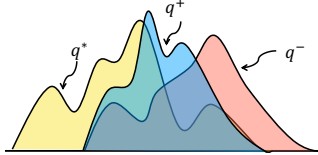

Figure 4: Sequential IRM projection of distributions, where $q^+ = \arg \min D_{\mathrm{KL}}(p||q^-) + D_{\mathrm{KL}}(p||q^*)$

$$R(q) = R(q^*) + \Delta q^T \nabla_q R(q^*)$$

we can compute the new distribution as

$$
\begin{aligned}
\Delta q^* \quad &= \quad \arg \min_{\Delta q} D_{\mathrm{KL}}(q^* + \Delta q || q^-) \\
\text{s.t.} \quad & \Delta q^T \nabla_q R(q^*) \leq \epsilon
\end{aligned}
$$

and then

$$p^+ = q^* + \Delta q^*$$

Or alternatively

$$
\begin{aligned}
p^*, q^* \quad &= \quad \arg \min_{p,q} D_{\mathrm{KL}}(p||q^-) + D_{\mathrm{KL}}(p||q) \\
\text{s.t.} \quad & \nabla_q R(q) = 0
\end{aligned}
$$

and then

$$q^+ = p^*$$

Or more simply

$$q^+ \quad = \quad \arg \min_p D_{\mathrm{KL}}(p||q^-) + D_{\mathrm{KL}}(p||q^*) \tag{22}$$

This last equation, shows how the new distribution is the intersection of the optimal distribution at the previous step $q^-$ and the current optimal distribution $q^*$. Fig 8 shows visually, how the new distribution is the result of projecting into two distributions $q^*$ and $q^-$.

## A.7 OUT OF DISTRIBUTION GENERALIZATION

The question arises if the property of generalization to out of distributions given by Theorem 9 in Arjovsky et al. (2019) also holds for BIRM and BVIRM.

**Lemma 9.** *If $\phi$ and $w$ are linear functions and $w \circ \phi = \Phi^T w$ is a solution of Eq.8 it then satisfies*

$$\Phi \mathbb{E}_{X^e} \left[ X^e X^{eT} \right] \Phi^T w = \Phi \mathbb{E}_{X^e, Y^e} \left[ X^e Y^{eT} \right]$$

*Proof.* Lemma 9 follows from the fact that

$$
\begin{aligned}
\nabla_w R^e(w \circ \phi) &= \Phi \mathbb{E}_{X^e} \left[ X^e X^{eT} \right] \Phi^T w - \Phi \mathbb{E}_{X^e, Y^e} \left[ X^e Y^{eT} \right] & (23) \\
&= 0 & (24)
\end{aligned}
$$

$\square$

For BIRM thus Theorem 9 of Arjovsky et al. (2019) applies directly. A similar results holds for the the BVIRM model

**Lemma 10.** *If $\phi \sim p_\phi$ and $w \sim p_\phi$ are linear functions and $w \circ \phi = \Phi^T w$ is a solution of Eq.9, with $\beta = 0$, it then satisfies*

$$\mathbb{E}_{\phi \sim q_\phi} \left\{ \Phi \mathbb{E}_{X^e} \left[ X^e X^{eT} \right] \Phi^T \right\} \bar{w} = \bar{\Phi} \mathbb{E}_{X^e, Y^e} \left[ X^e Y^{eT} \right]$$

*where $\bar{\Phi} = \mathbb{E}_{\Phi \sim p_\phi}[\Phi]$ and $\bar{w} = \mathbb{E}_{w \sim p_w}[w]$ are the mean values.*

*Proof.* Lemma 10 follows from the fact that

$$
\begin{aligned}
\nabla_{q_w} Q_w^e(q_w, q_\phi)|_{\beta=0} &= \nabla_{q_w} \mathbb{E}_{\substack{w \sim q_w \\ \phi \sim q_\phi}} R^e(w \circ \phi) & (25) \\
&= 0 & (26)
\end{aligned}
$$

We now take the Fréchet directional derivative in the $\eta$ direction that is the limit of

$$\delta_{q_w, \eta} \mathbb{E}_{\substack{w \sim q_w \\ \phi \sim q_\phi}} R^e(w \circ \phi) = \lim_{\epsilon \to 0} \frac{1}{\epsilon} (\mathbb{E}_{\substack{w \sim q_w \\ \phi \sim q_\phi}} R^e(((w + \epsilon \eta) \circ \phi) - \mathbb{E}_{\substack{w \sim q_w \\ \phi \sim q_\phi}} R^e(w \circ \phi))$$

which is obtained when we differentiate the distribution $q_w \to q_w + \epsilon \eta$. Since $\delta_{q_w, \eta} \mathbb{E}_{w \sim q_w \phi \sim q_\phi} R^e(w \circ \phi) = \mathbb{E}_{w \sim q_w \phi \sim q_\phi} 2\eta^T \Phi \mathbb{E}_{X^e} \left[ X^e X^{eT} \right] \Phi^T w - 2\eta^T \Phi \mathbb{E}_{X^e, Y^e} \left[ X^e Y^{eT} \right]$ we can factorize for the direction $\eta$ and obtain

$$\delta_{q_w} R^e(w \circ \phi) = 2 \mathbb{E}_{\substack{w \sim q_w \\ \phi \sim q_\phi}} \Phi \mathbb{E}_{X^e} \left[ X^e X^{eT} \right] \Phi^T w - \Phi \mathbb{E}_{X^e, Y^e} \left[ X^e Y^{eT} \right]$$

We can now derive the Lemma by requiring $\delta_{q_w} R^e(w \circ \phi) = 0$ $\square$

Theorem 9 of Arjovsky et al. (2019) now holds when $\phi$ has rank $r > 0$ in expectation with respect to the invariant distribution $q_\phi$, i.e. $\mathbb{E}_{\phi \sim q_\phi} \text{rank}(\Phi) = r$.

## A.8 GENERALIZED ADMM

The following generalization of ADMM holds:

**Lemma 11** (GADMM). *Suppose we want to minimized*

$$\min_x \quad \sum_i f_i(x) | g_i(x) = 0, \forall i \in I \tag{27}$$

*we can equivalently solve the following problem*

$$\min_{x_i, z} \quad \sum_i f_i(x_i) | x_i = z, g_i(x_i) = 0, \forall i \in I \tag{28}$$

*using the following update role (scaled ADMM)*

$$x_i^+ \quad = \quad \arg\min_{x_i} L_\rho(x_i, x_{-i}^-, u_i^-, z^-, v_i^-), \forall i \in I \tag{29a}$$

$$z^+ \quad = \quad 1/N \sum_i (x_i + u_i) \tag{29b}$$

$$u_i^+ \quad = \quad u_i^- + (x_i^+ - z^+) \tag{29c}$$

$$v_i^+ \quad = \quad v_i^- + g_i(x_i^+) \tag{29d}$$

*where the augmented Lagrangian*

$$L_\rho(x_i, u_i, z, v_i) \quad = \quad \sum_i f_i(x_i) + \frac{\rho_0}{2} \sum_i \|x_i - z + u_i\|^2 + \frac{\rho_1}{2} \sum_i \|g_i(x_i) + v_i\|^2 \tag{30}$$

*and $x_{-i} = \{x_j, j \neq i\}$ is the set of all other variable, expect the $i$-th.*

## A.9 CONTINUAL VARIATIONAL INFERENCE

Following Nguyen et al. (2018) we can state the following lemma.

**Lemma 12** (Variational Continual Learning). *Suppose we have a sequence of datasets $D_i, i = 1, \ldots, t$ drown i.i.d, then the variational estimation of the distribution $q_t$ at step $t$ is given as projection on KL divergence*

$$q_t(\theta) = \arg\min_{q(\theta)} D_{\mathrm{KL}}\left( q(\theta) \| \frac{1}{Z_t} q_{t-1}(\theta) p(D_t|\theta) \right)$$

*with $Z_t = \int q_{t-1}(\theta) p(D_t|\theta) d\theta$ the normalization factor, which does not depends on $q$.*

*Proof of Lemma 12.* Let denote $D_1^t = \bigcup_{i=1}^t D_i$, from i.i.d. $p(D_1^t) = \prod_{i=1}^t p(D_i)$. We are interested to maximase the a posteriori probability of the paramters give the data $p(\theta|D_1^t)$

$$p(\theta|D_1^t) = \frac{1}{p(D_t)} p(\theta|D_1^{t-1}) p(D_t|\theta)$$

since

$$
\begin{aligned}
p(\theta, D_1^t) \quad &= \quad p(\theta|D_1^t) p(D_1^t) \\
&= \quad p(\theta) p(D_1^t|\theta) \\
&= \quad p(\theta) \prod_{i=1}^t p(D_i|\theta) \\
&= \quad p(\theta) p(D_1^{t-1}|\theta) p(D_t|\theta) \\
&= \quad p(\theta, D_1^{t-1}) p(D_t|\theta) \\
&= \quad p(\theta|D_1^{t-1}) p(D_1^{t-1}) p(D_t|\theta)
\end{aligned}
$$

thus

$$
\begin{aligned}
p(\theta|D_1^t) \quad &= \quad \frac{1}{p(D_1^t)} p(\theta|D_1^{t-1}) p(D_1^{t-1}) p(D_t|\theta) \\
&= \quad \frac{p(D_1^{t-1})}{p(D_1^t)} p(\theta|D_1^{t-1}) p(D_t|\theta) \\
&= \quad \frac{1}{p(D_t)} p(\theta|D_1^{t-1}) p(D_t|\theta)
\end{aligned}
$$

We now use a probability distribution which approximates the distribution at step $t - 1$

$$q_{t-1}(\theta) \approx p(\theta|D_1^{t-1})$$

when then want to approximate at time $t$

$$q_t(\theta) \approx \frac{1}{p(D_t)} q_{t-1}(\theta) p(D_t|\theta)$$

This can be obtain by minimizing the KL divergence of the variational distribution $q_t$ and the distribution induced by the previous step approximation, thus

$$q_t(\theta) = \arg\min_{q(\theta)} D_{\mathrm{KL}}\left(q(\theta)||\frac{1}{Z_t}q_{t-1}(\theta)p(D_t|\theta)\right)$$

$\square$

**Lemma 13** (VCLv2). *The minimization of the VCL defined in Lemma 12, is equivalent to solve the following minimization*

$$q_t(\theta) = \arg\max_{q(\theta)} \mathbb{E}_{\theta\sim q(\theta)}\{\log p(D_t|\theta)\} - D_{\mathrm{KL}}(q(\theta)||q_{t-1}(\theta))$$

*with $N_t$ i.i.d. samples*

$$\mathbb{E}_{\theta\sim q(\theta)}\{\log p(D_t|\theta)\} \quad = \quad \frac{1}{N_t}\sum_{i=1}^{N_t} E_{\theta\sim q(\theta)}\{\log p(y_i^t|\theta, x_t^t)\}$$

Where the second term can be computed in closed form for known distribution as for example with the Gaussian distributions, whereas the expectation can be approximated by Monte Carlo sampling. For a general loss function we can substitute the reconstruction probability with the loss function associated with a neural network parametrized by $\theta$

$$\log p(y_i^t|\theta, x_t^t) \leftarrow \ell(y_i^t, (w \circ \phi)_\theta(x_t^t))$$

$$\mathbb{E}_{\theta\sim q(\theta)}\{\log p(D_t|\theta)\} \quad \leftarrow \quad \frac{1}{N_t}\sum_{i=1}^{N_t} E_{\theta\sim q(\theta)}\ell(y_i^t, (w \circ \phi)_\theta(x_t^t))\}$$

*Proof of Lemma 13.* The Lamma follows from the definition of the KL diveregnce

$$
\begin{aligned}
D_{\mathrm{KL}}\left(q(\theta)||\frac{1}{Z_t}q_{t-1}(\theta)p(D_t|\theta)\right) &= \mathbb{E}_q(\ln q(\theta) - \ln q_{t-1}(\theta) - \ln p(D_t|\theta) + \ln Z_t) \\
&= \mathbb{E}_q(\ln q(\theta) - \ln q_{t-1}(\theta)) - \mathbb{E}_q \ln p(D_t|\theta) + \mathbb{E}_q \ln Z_t \\
&= D_{\mathrm{KL}}(q(\theta)||q_{t-1}(\theta)) - \mathbb{E}_q \ln p(D_t|\theta) + \ln Z_t
\end{aligned}
$$

The last term does not depend on $q$. Thus the result follows. $\square$

If we substitute the log of the posterior probability with a specific loss function we obtain the following Corollary.

**Corollary 14** (Continual Variational Bayesian Inference). *Given a loss function $\ell(y, \hat{y})$, the variational continual learning is formulated as*

$$q_t(\theta) = \arg\min_{q(\theta)} \mathbb{E}_{(x,y)\sim D_t}\mathbb{E}_{\theta\sim q(\theta)}\{\ell(y, f_\theta(x))\} + D_{\mathrm{KL}}(q(\theta)||q_{t-1}(\theta)), \tag{31}$$

*with $f_\theta = (w \circ \phi)_\theta$*

### A.10  PROOFS

*Proof of Theorem 3.* Let first first recall that $D_{\mathrm{KL}}(p||q) = \int p(x) \ln \frac{p(x)}{q(x)} dx$. If $q(x) = 0$ then $p(x) = 0$ otherwise the distance is infinite. Second if $p(x) = 0$, then the contribution of $q(x)$ is not considered since the integral is taken of the support of $p$, thus, since the intersection is not null and $p$ is the result of an optimization, the support of $p$ is the intersection of the support of $q$ and the support of $P$. $\square$

### A.11  DATASETS AND COLOR CORRECTION

We here visualize few of the dataset and color correlations. Figure 5 shows Fashion-MNIST and the b11 color correlation. In the test environment the background color of each class is inverted. In Figure 6 we show the dataset as generated from Ahuja et al. (2020). In Figure 7 we show the EMINST (letter) and KMNIST dataset.

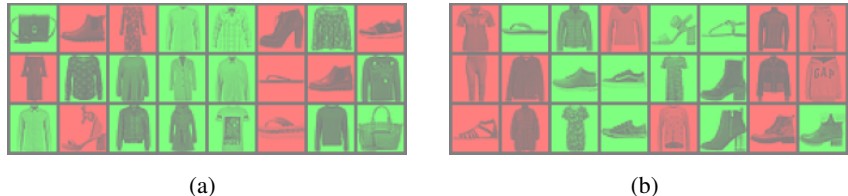

Figure 5: Fashion MNIST dataset training (a) and testing (b) environments; the color is inverted based on the b11 color correlation scheme, where the background color depends on the class of the image. In the test environment the dependency is inverted.

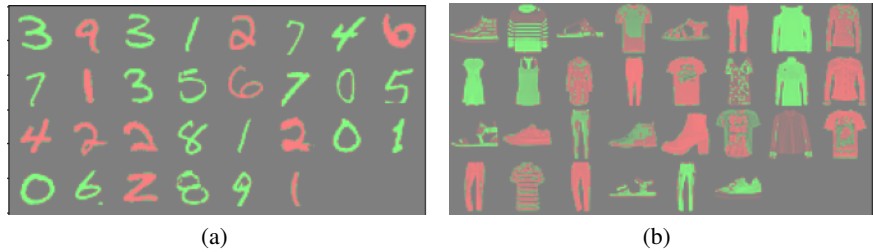

Figure 6: MNIST dataset (a) and Fashion MNIST (b) environments as defined in Ahuja et al. (2020)

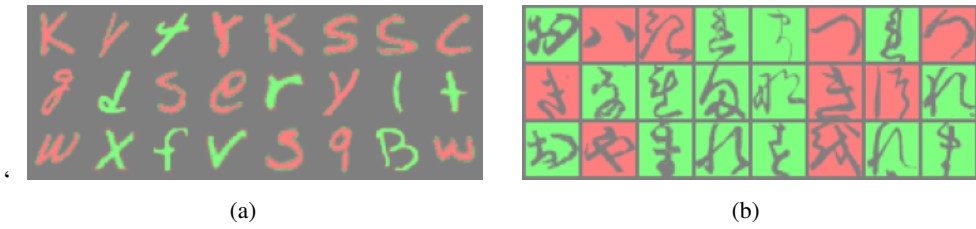

Figure 7: Examples of the EMNIST dataset (a) and of the KMNIST (b).

## A.12   HYPER-PARAMETER SEARCH AND EXPERIMENTAL SETUP

We performed hyper-parameter search around the suggested values from the original works and the values selected based on the best performance on the test environment. To implement a complete comparison we used for training $1'000$ samples randomly drawn from each environment. All methods were trained on the same data, using random seed reset. We trained all method with 100 epochs on a batch size of 256.

- **IRM**: $\gamma = 91257$, threshold = 1/2 epochs, learning rate $2.5e^{-4}$
- **IRMG**: warm start=300, termination accuracy 0.6, learning rate $2.5e^{-4}$, dropout probability 75%, weight decays = .00125
- **ERM**: learning rate $1e^{-3}$, dropout probability 75%, weight decays = .00125
- **MER**: memory size 100 (10% of the samples), learning rate $1e^{-3}$, replay batch size =5, $\beta = .03$, $\gamma = 1.0$
- **GEM**: memory size 100 (10% of the samples), learning rate $1e^{-3}$
- **EWC**: memory size 100 (10% of the samples), learning rate $1e^{-3}$, regularization 0.1
- **VCL,VCLC**: learning rate = $5e^{-3}$, corset size 100 (10% of the samples),
- **C-BVIRM, C-VIRMG, C-VIRMv1**: weight decays = .00125, $\beta = 1.$, number evaluations 5, $\rho_0 = \rho_1 = 10$, step threshold =1/2 epochs, $\delta\rho = 100$, learning rate $1e^{-3}$

The neural network architecture is composed of 2 non-linear Exponential Linear Unit (ELU) activated Full Connected layers of size 100, followed by a linear full connected layer. Each layer with dropout. Dropout is not present in VCL/VCLC since not implemented in the original work. Training loss is the Cross Entropy. We tested also with the feature extraction layer separated, but with no advantage, since the test set-up only consist of one task.

The IRMv1, IRMG and ERM methods, similarly to the other methods, are trained sequentially as data from each new environment arrives. The Continual Learning methods are allowed to have a limited memory of samples from previous environments.

## A.13   HYPER-PARAMETERS OF ENVIRONMENT INFERENCE FOR CONTINUAL INVARIANT LEARNING

We list below the values of hyper-parameters in EIIL for continual invariant learning:
2 layers,
390 hidden neurons,
501 epochs,
l2 regularizer weight: 0.00110794568,
learning rate: 0.0004898536566546834,
numbber of runs: 10,
number of EIIL iterations: $10'000$,
number Monte Carlo evaluations: 3,
penalty anneal iterations: 190,
penalty weight: 191257.18613115902,
prior weight: $1e^{-6}$

### A.13.1   SYNTHETIC DATASET

The Synthetic Dataset is described in (Arjovsky et al. (2019)) for testing IRM and it is defined by a Structural Causal Model (Pearl (2009)), where a variable $y \in \mathbb{R}^N$ is generated by $x_1 \in \mathbb{R}^N$, while $x_2 \in \mathbb{R}^N$ is generated by $y$. The observed variable is $x = (x_1, x_2)$. The structural equations are

$$x_1 = \epsilon_1,\ \epsilon_1 \sim N(0, \sigma_1^2) \tag{32}$$

$$y = x_1 + \epsilon_y,\ \epsilon_y \sim N(0, \sigma_e^2) \tag{33}$$

$$x_1 = y + \epsilon_2,\ \epsilon_2 \sim N(0, 1) \tag{34}$$

with $\sigma_1$ fixed and $\sigma_e$ dependent on the environment. We compared with ERM, IRM (Arjovsky et al. (2019)), IPC (Invariant Prediction), which is the method proposed in Peters et al. (2015), and EIIL (

Table 5: Mean accuracy (over 5 runs) on Synthetic Dataset (Arjovsky et al. (2019),Creager et al. (2020)). BIRM refers to our bilevel objective Eq. (8) optimized with ADMM.

|  | Causal MSE | Noncausal MSE |
|---|---|---|
| ERM | $0.827 \pm 0.016$ | $0.824 \pm 0.015$ |
| ICP | $1.000 \pm 0.000$ | $0.756 \pm 0.423$ |
| IRM | $0.280 \pm 0.006$ | $0.290 \pm 0.009$ |
| **BIRM** | $0.183 \pm 0.005$ | $\mathbf{0.184} \pm 0.002$ |
| EIIL(IRM) | $\mathbf{0.180} \pm 0.026$ | $0.188 \pm 0.033$ |

Creager et al. (2020)). We use a similar set up of Creager et al. (2020), with $N = 4$. The invariant model is given by $w = (1, 0)$.

