# OpenReview forum: "Continual Invariant Risk Minimization"
_ICLR.cc/2021/Conference — Reject_

### Official Review · AnonReviewer1 · 2020-10-27
**The experimental results do not pass basic sanity checks**

**Rating:** 3
**Confidence:** 4

**Review:**

# Post-discussion update

The authors have significantly updated the paper during the discussion period. I have seen the changes, but they unfortunately do not substantiate the claim that the proposed methods are learning anything meaningful.

In the new results in table 1, the train accuracy now is better than random, but the test accuracy on the unseen environments is worse than a model that makes uniformly random predictions. This implies that the proposed method is not learning to ignore the spurious features at all. It might seem from table 1 that C-VIRMv1 is performing well --- it has 46% accuracy on the test environment after all. However, C-VIRMv1 also has the worse performance on the train set (50% accuracy). In-fact, the test or train performance alone does not mean anything in this benchmark. The goal of the benchmark is to do well on the train set while also generalizing to an unseen environment. Doing well on test set by doing poorly on the training set is not progress.

The new results in table 3 are equally troubling. The authors claim that the results in table 3 show that C-VIRM with EIIL is performing better than IRM with EIIL, but the data does not support the claim. Both IRM and C-VIRM are performing similarly; all the results are with-in error margins of each others, and the table can not be used to make any claims.

I would encourage the authors to do a more systematic study of the proposed method. Investigate if the proposed methods are learning to ignore the spurious correlation at all (Table 1 suggests they are not) and only make claims that are supported by the data. In its current form, I cannot recommend this paper for acceptance.

# Initial review

I'm going to keep the initial review short to point out a fundamental issue with the paper. Until this issue is addressed, I see little point in discussing the paper at length. Hopefully the authors can address the issue in the discussion period and I will update my review to include other aspects of the paper.
## Issue No 1 ##
 The proposed solution is arguably not learning at all. Both MNIST and FashionMNIST, as used in this paper and earlier IRM papers, are binary classification benchmarks (Five classes have label 0 and remaining five have label 1). A completely randomly initialized neural network gets ~50% accuracy on a binary classification task. This is not just speculation, I have run experiments using a randomly initialized network on these benchmarks to confirm that they do indeed get ~50% +- 2% accuracy on both train and test set. Coincidently, both C-BVIRM and C-VIRMG also get ~50% accuracy on the train/test set. This would imply that the model learned by C-BVIRM and C-VIRMG is indistinguishable from a model that does no learning. Here are some other baselines that will do as well as C-BVIRM:

1. Set learning rate of ERM to be infinitesimally small so that no learning happens in the given number of steps (set learning rate to $e^{-100}$ for instance).

2. Make completely random changes in the weight instead of updating the weights using gradients.

3. Make no changes to the network.

Clearly 1, 2 and 3 are bad learning algorithms (the first and third are not learning at all whereas the second is making arbitrary updates), yet all three of them would perform as well as the two algorithms proposed in the paper.

Does this mean the two methods proposed in the paper are bad? Not necessarily. The BIRM formulation of the problem seems technically correct and the authors are tackling and important and interesting problem. However, the poor performance of C-BVIRM does imply that the authors need to look at their experiment results in more detail and do some sanity checks.

A sanity check for checking if C-BVIRM is learning anything at all is to test it on a benchmark for which both train and test environments have a constant $p_c$ (say 0.1). A good learner should get 90% accuracy on both train and test environment in such a benchmark (Both IRM and ERM would get 90% accuracy). My guess is that C-BVIRM and C-VIRMG would get 50% accuracy on train/test when trained using the same parameters as used to run the experiments in the paper.

## Issue No 2 ##
Please see the private comment for the second issue.

Until issue 1 is addressed, discussing the technical details of the paper wouldn't make a difference and I can not recommend an acceptance.

---

> ### Author Response · Authors · 2020-11-18
> **Thank you for your review - Sanity check**
>
> Dear Reviewer,
>
> Thank you for your helpful feedback. We executed the sanity check that you suggest, where test and training environments’ image colours are drawn with probability pc=0.1. We considered n=2 environments for training and one for testing, where samples are MNIST images with colour correction b01. The rest of the experiment is described in the paper. As the following table (Table A) shows (averaged over 5 seeds), the learned models do not behave randomly.
>
>
> Table A
>
> | Method     | Train Accuracy   |   Test Accuracy |
> | -------------     |:-------------:    | :-------------:   |
> | C-BVIRM          | 64.6% (4.3%)      | 61.1% (4.9%)      |
> | C-VIRMG          | 69.7% (4.0%)      | 66.0% (2.3%)      |
> | C-VIRMv1         | 59.6% (3.4%)      | 59.2% (3.2%)     |

---

> > ### Comment · AnonReviewer1 · 2020-11-23
> > **Thanks for running the sanity check**
> >
> > Thanks for running the sanity check. While the results are not 50%, they are closer to 50% than 90% indicating that C-BVIRM, C-VIRMG, and C-VIRMv1 are indeed heavily regularizing learning. For instance, for pc=0.1 on train and test environment, ERM would learn to achieve ~ 90% accuracy on train and test environments very quickly.
> >
> > I'm still not sure why 50% accuracy on a binary classification task is meaningful at all as a randomly initialized network would achieve similar results. Can the authors provide any justification?

---

> > > ### Author Response · Authors · 2020-11-25
> > > **Thank you for your feedback**
> > >
> > > Thanks for your question. Your comment has really helped us to improve the experimental setup.
> > >
> > > In order to strengthen our accuracy, we made the following changes:
> > > 1. We selected models and hyper-parameters much more carefully. Specifically, we removed the drop-out, originally proposed in (Ahuja et al., 2020), while training BVIRM methods. This was the result of selecting the hyper-parameters based on the performance on the test environment.  For the new simulations, we selected hyper-parameters that show better convergence in the training set.
> > > 2.  We replaced "ELU" activation function with "ReLU". This was observed beneficial in the new experiments, when comparing with (Creager et al., 2020). Interestingly there was not issues for computing the second derivative, required in the IRM constraint.
> > > 3. We cleaned the code and identified small differences in the training loop of ADMM (which may influence the performance of our C-BVIRM) and fixed that. We plan to make code publicly available upon acceptance.
> > >
> > > We clarified experimental setup in the revised submission. We also updated quantitative results in Table 1 and Table 2, which suggest that our method offers a more reasonable performance gain over competitors. As can be seen, the accuracy values are different from 50%.
> > >
> > > Further, based on suggestions from Reviewer 4, we noticed that, the performance of our method can be improved when it is combined with Environment Inference for Invariant Learning (EIIL) (Creager et al., 2020), a recently proposed strategy of automatic environment partition. These results are shown in Tables 3 and 4 in the revised submission.

---

### Official Review · AnonReviewer4 · 2020-10-27
**Paper proposes an extension of IRM for continual learning. The problem is interesting, the method proposed is natural and makes sense. Theoretical justifications are incomplete and a bit misleading. Experiments are below par and need a lot of work.**

**Rating:** 5
**Confidence:** 5

**Review:**

Summary:
In this work, the authors consider the problem of continual learning with distribution shifts.  The work extends the recent work invariant risk minimization (IRM) from Arjovsky et al. to a continual learning setup. IRM was designed as an offline learning framework. In this work, the authors consider the setting where the different domains arrive sequentially. The authors propose a Bayesian extension of the IRM framework that allows sequential updates as the environments arrive. They provide a justification for how the KL divergence term helps in shrinking the support continually to arrive at an approximately invariant support. Several experiments were carried out on colored MNIST and its variants to show how the proposed scheme is better than existing continual learning methods and existing IRM frameworks.

Pros:
I like the problem the authors consider. It is indeed important to understand how to extend IRM type frameworks in settings such as continual learning. I also liked the ADMM based approach taken by the authors as this approach gives another way of learning IRM based predictors in addition to existing approaches.

Cons:

There are several problems with the paper. I discuss these problems below in different subsections. If these major issues are addressed in the rebuttal, I would be open to changing my score.

a) Information Projection (Theorem 3)
In the work, the authors show in Theorem 3 as to how when the model is trained sequentially the support of the distribution shrinks.

Let us consider optimization in 2 time steps

t=1, q_{t} = min_{q_{t}\in P} KL (q_t||q_{t-1}) --> support of q_1 = P \cap support of q_0

t=2, q_{t} = min_{q_{t}\in P} KL (q_t||q_{t-1}) --> support of q_2 = P \cap support of q_1 = P \cap P \cap support of q_0 = P \cap support of q_0

It is not clear from just the KL term why support would strictly shrink. The reason why the authors approach works is because the first term based on the IRM loss allows to update and shrink the support and the KL divergence term ensures that there is no need to expand the existing support. Define the first term in the loss associated with IRMv1 as Q_{IRMV1}(q_t)

t=1, q_{t} = min_{q_{t}\in P} Q_{IRMv1}(q_t) + KL (q_t||q_{t-1}) --> support of q_1 \subset P \cap support of q_0

t=2, q_{t} = min_{q_{t}\in P} Q_{IRMv1}(q_t)  + KL (q_t||q_{t-1}) --> support of q_2 \subset P \cap support of q_1

It is the combination of the IRM term with KL term that ensures shrinkage and KL term on its own only ensures support does not expand. A proof or an illustration of what I state would have been nice as the current intuition from the authors is incomplete and to some extent a bit misleading.

b) Why penalize the IRM constraints as well?
The authors arrived at a variational formulation in 9a and 9b with new objective functions defined in 10 a and 10b. It is unclear why it was chosen to impose KL penalty in the constraint (10b) as well. If not a theoretical justification, an experiment in the supplement illustrating why these choices were made would be nice. At least some good intuition needs to be provided.

c) About the ADMM approach:
I am happy to see the authors tried the ADMM approach. However, one thing that is unclear is it does not necessarily serve the purpose of solving the problem authors want. From the experiments it seems a continual extension of IRMv1 itself suffices. Is there any other reason why ADMM approach was used. Also, if ADMM approach was proposed as another way to solve standard                                                                                                               IRM in the offline setting, then some experiments explaining any advantage would be useful. At least some offline comparisons would make a better case for using ADMM.

d) Why not empirically compare with Javed et al.?

As the authors correctly identify there is a recent work from Javed et al., which solves the same problem but makes more assumptions about the data.  I understand that the work uses one hot encoding of color etc. Despite that you can allow your model to also have access to the same data format and then see if your model has any more advantage to offer over and above what was done in Javed et al.

e) Comparisons with offline IRM based methods (IRMv1 Arjovsky et al., IRMG Ahuja et al.) do not seem to have been carried out in a fair manner.
The authors have not provided a very clear description of how IRM based methods were trained. It seems to me that the authors are running offline IRM based methods in the following manner.  They provide the data from the first environment and then keep the offline IRM based models fixed and not update them when the data from the next environment arrives. If they indeed update the offline IRM based methods, then there is no reason why the methods perform so poorly.  What was the reason for not updating these models? The reason I am suspecting is that everytime retraining the entire model on the entire data is computationally expensive. If that is the concern, then authors should have shown a comparison of run-times. The authors should have run experiments allowing all methods the same run-times and then shown that IRM based methods are still not able to perform better.
For instance if the time to train continual IRM based model proposed by authors for the two environments is a total of 20 seconds.
Suppose the total time to train a standard IRMv1 is say a total of 15 seconds for the first environment. There is still a five second window for the IRMv1 model to be updated.  If run-time comparisons are non-trivial, then at least there should be a budget on the total number of gradient computations that is fixed across methods.


f) Compare with Creager et al.

The reason IRM based offline methods did not perform well is because they were given data from one environment only.  Recently, in Creager et al., it was shown how one can learn to split the data to create new environments and then train an IRM on those environments.  The authors should compare with the Creager et al. as follows. When the data from first environment comes in you can use the approach introduced in the paper http://www.gatsby.ucl.ac.uk/~balaji/udl2020/accepted-papers/UDL2020-paper-045.pdf to learn environments (code available at https://github.com/ecreager/eiil). Therefore, by doing this you can split the data in the first environment into smaller environments and use IRM. This would lead to a lot of performance improvement for the offline IRM based methods.


g) Current experiments do not reflect the true potential of the method proposed by the authors:

The current experiments do not do a good job of reflecting a continual learning setup. Simple modifications on existing IRM based methods can outperform the method proposed by the authors. However, I believe the method proposed by the authors is nice and much more experimentation is needed. I make some suggestions on how to improve the experiments.

It seems that in the current experiments after 2 environments there is no more gain from using any method, in fact more environments seem to hurt. I would encourage the authors to do an experiment where adding more environments actually removes spurious correlations more and more and helps.  To this end, there are three suggestions I would like to make:

i) First let us understand is why is two environment sufficing in colored MNIST dataset.  The reason is that the dimension of the spurious factor, i.e., color is small (only two colors). You can increase the number of colors, then more environments will be needed to decorrelate.

ii) If  adding colors and doing comparisons using raw images is hard, then you can try using the one hot encoded colored MNIST from Javed et al. and add more colors to it.

iii) Another suggestion would be to try the regression experiments from Arjovsky. In Arjovsky, the authors use two environments only. However, the two environments differ a lot 0.2 and 2.0 variance. If you introduce a sequence of environments from 0.2 to 2.0, say 0.2, 0.4, 0.6, ...2.0., then it is likely that more environments will have some benefit.

Quality: The solution proposed seems promising. There are problems with theoretical justification. Experiments need a lot of work as I stated above.

Significance: The problem considered in the paper is interesting. The experiments carried out do not do a justice to the problem being considered.

Clarity: The writing of the paper can be improved. The experiments need to be described much more clearly and use the space in the supplement to give all the details. Also, the steps of ADMM shoould have been better explained.




References

Javed et al. "Learning causal models online"

Creager et al. "Environment Inference for Invariant Learning"

---

> ### Author Response · Authors · 2020-11-22
> **Thank you for your review - Reply to your Questions a, b, c and d**
>
> We appreciate the comprehensive evaluation and valuable questions from reviewers. Please let us first address your questions a, b,c, d and e.
>
> [Qa] Information Projection (Theorem 3) In the work, the authors show in Theorem 3 as to how when the model is trained sequentially the support of the distribution shrinks. [...]
>
> [Ra] Thank you for your comment. Our description was not complete. Theorem 3 states a general property, while the sequential projection was not completely clarified and we also did not give a formal connection with the variational model.
> Now, we updated Figure 1, where the missing set has created inconsistency. We additionally provide in supplementary material A.6 how information projection is related to IRM objective, which suggests that it is the combination of IRM and KL divergence that ensures the shrink of the support. Lemma 8 provides the property of sequential information projection to formalize what the reviewer commented on.
>
> [Qb] Why penalize the IRM constraints as well? [...]
>
> [Rb] We added in the supplementary material Lemma 7 that addresses this point. The intuition is that the second equation is associated with the optimality condition on all environments on the classifier (predictor) parameter $w$ that for the variational case is in the distribution $q_w$. The loss function is the lower bound (ELBO) that contains both the error term and the divergence with respect to the prior distribution, which for $w$ is $p_w$, while the parameters of the function $\phi$ are not used. In other words, the intuition of having a KL term in Eq.10b is that the predictor parameters’ distribution  ($q_w$) shall be optimal in all environments, where the loss function includes the KL term. We also notice that a term in the Eq.10a was missing. Thank you for the question.
>
> [Qc] About the ADMM approach: I am happy to see the authors tried the ADMM approach. However, one thing that is unclear is it does not necessarily serve the purpose of solving the problem authors want. From the experiments it seems a continual extension of IRMv1 itself suffices. Is there any other reason why ADMM approach was used. Also, if ADMM approach was proposed as another way to solve standard IRM in the offline setting, then some experiments explaining any advantage would be useful. At least some offline comparisons would make a better case for using ADMM.
>
> [Rc] Thank you for the observation. ADMM provides a framework for solving BIRM and BVIRM. The advantage of ADMM is for solving in parallel multiple environments for the off-line training. More specifically Eq.11b requires synchronization among parallel environments, but Eq.11a,Eq.11c and Eq.11d can be computed in parallel. This is also now described in section 3.4.
> In the online setting, the performances of the continual version of IRMv1 are indeed very close. However, our ADMM still provides an effective alternative and enriches our understanding of the optimization. In this sense, we feel that our ADMM also contributes a lot to the IRM and its extensions.
>
> [Qd] Why not empirically compare with Javed et al.? [...]
>
> [Rd] We did not compare with Javed since the setup is different. We would have needed to modify our approach to have a fair comparison. Further we impose a budget in the training, while the Javed approach is based on reinforcement learning and long training, that would be difficult to compare with our experimental set up.
>
> [Qe] Comparisons with offline IRM based methods (IRMv1 Arjovsky et al., IRMG Ahuja et al.) do not seem to have been carried out in a fair manner. [...]
>
> [Re] Thank you for the your question. We did not stop the training of IRMG and IRMv1. We trained all methods with the same set up. The offline IRMv1 and IRMG do not consider sequential setting, so we expected to overfit in the training. The IRMv1 and IRMG are presented with sequential environments and continuously training as new data is presented. Since they see only one environment at time it would not be possible to identify invariant models. We show this effect experimentally, but it is not a criticism of these methods. Rather, it is simply that the sequential environment breaks the assumption upon which these methods have been built.
> We added in supplementary material the description on how IRMv1 and IRMG are trained. We do not report computational times, since these may vary with implementation, even if in general Variational methods require more computation because we learn a distribution of networks (versus a single network) and we use Monte Carlo sampling for training.

---

> > ### Comment · AnonReviewer4 · 2020-11-24
> > **Response**
> >
> > Thank you for your efforts to revise and provide responses. In [Re] you say IRM cannot work with one environment data. I have pointed in f), the reference to address exactly this issue. So currently a fair comparison with most important approach IRM is not done.

---

> > > ### Author Response · Authors · 2020-11-25
> > > **Thank you for your review - Comparison with Creager et al**
> > >
> > > Thanks for your suggestion on EIIL [Creager et al. (2020)].
> > >
> > > We evaluate and explore EIIL from two aspects.
> > >
> > > First, we begin with the same synthetic regression dataset in the original IRM (Arjovsky et al., 2019). In case of only one environment data, the following table (see also Section A.13.1 and Table 5 in the revised submission) suggests that our BIRM (i.e., Eq. (8) optimized with ADMM) performs similarly to IRM + EIIL.
> > >
> > > |        Method                   |     Causal MSE                 |            Noncausal MSE |
> > > | -------|:--------------:|:-------------:|
> > > | ERM                |        0.827 (0.016)               |                  0.824 (0.015) |
> > > | ICP                   |       1.000 (0)                      |                 0.756 (0.423) |
> > > | IRM                  |       0.280 (0.006)                |                  0.290 (0.009) |
> > > | BIRM                |       0.183 (0.005)               |                   0.184 (0.002) |
> > > | EIIL (IRM)        |       0.180 (0.026)                |                  0.188 (0.033) |
> > >
> > > Second, motivated by the experimental setup in (Creager et al., 2020), we generalize our continual invariant learning models into an environment-agnostic setting by integrating ELLI into our proposed C-VIRMv1. We do notice an obvious performance gain of our C-VIRMv1 and IRMv1 when they are combined with EIIL. The new results in Tables 3 and 4 (Section 5.3) of the revised submission further strengthened the argument that "inferring environments directly from observational data has the potential to improve (continual) invariant learning relative to using the hand-crafted environments". Moreover, it seems C-VIRMv1+EIIL always outperforms IRMv1+EIIL.

---

> > > > ### Comment · AnonReviewer4 · 2020-11-25
> > > > **Response**
> > > >
> > > > Thank you for running the comparison. In all the comparisons that you do when you present data from one environment only to the original IRMv1, then the only fair comparison is to use EIIL as well on top. Now after seeing your experiments it seems from Table 4 that IRMv1 with EIIL and C-VIRMv1 with EIIL are similar in performance. This was my original belief that it is important to show gains w.r.t simple baseline such as EIIL + IRMv1.
> > > > In light of the new comparisons, my suggestion to the authors would be:
> > > > 1.If the method can be improved further so that the gains can be more pronounced (currently everything is in the same ball park).
> > > > 2.  I feel this paper will have a good potential if more convincing experiments that reflect a continual learning setup are done. I believe in the current experiments IRMv1 with EIIL is sufficient. However, I believe the method proposed by the authors can have benefits provided right type of experiments are done (See point g) in my original response).

---

> > > > > ### Author Response · Authors · 2020-11-25
> > > > > **Thank you for your insightful comments**
> > > > >
> > > > > Thank you again for your insightful comments,
> > > > >
> > > > > Indeed, from Table 4, the performance of IRMv1 is similar to C-VIRMv1. However, we also observed a constant performance gain of C-VIRMv1 + EIIL over IRMv1 + EIIL. In our Table 2, we also compared with benchmark continual learning methods, including EWC, GEM, MER and VCL. Results suggest that our methods always have better test accuracy, especially for C-VIRMv1.
> > > > >
> > > > > In our paper, we highlighted a limitation of current methods, and also based on your intuition (thank you for that) we delineated few directions to address this problem: 1) modelling the invariant models as distributions and discover invariant distributions in sequential environments by sequential information projection and 2) infer invariant models from single dataset by dataset partition (as EIIL provides). Indeed the second approach still has the problem of caring over the learned invariant model into future datasets. Further, as highlighted in the experiments on sample size (Table 4) in the new experimental section, when the number of samples is too small, even environment inference does not help in discovering invariant models.
> > > > >
> > > > > Regarding your point (g), we thank you again for the insightful suggestions, which we already partially incorporated in the current work, and we will consider them for future extensions.

---

### Official Review · AnonReviewer3 · 2020-10-29
**An interesting extension of IRM to the continual learning setting**

**Rating:** 6
**Confidence:** 4

**Review:**

This paper extends the idea of invariant risk minimization (IRM) initially introduced by Arjovsky et al. (2019) to the setting of continual learning in which environments are observed sequentially rather than concurrently. This extension is implemented under a variational Bayesian and bilevel framework and the optimization is solved using a variant of the alternating direction method of multiplier (ADMM). The authors demonstrate the superiority of the proposed methods on variants of Colored MNIST.

Pros:
+ The proposed method is a natural and practical extension of IRM and IRMG, which I believe is more applicable to many real-world scenarios.
+ The resulting bilevel problem can be efficiently addressed using the alternating direction method of multipliers.

Concerns:
- My main concern is about the generalization theory in this continual setting. In both IRM and IRMG, the authors provide the conditions under which the OOD generalization can be guaranteed, as stated in Assumption 8 and Theorem 9 of Arjovsky et al. (2019) and in Assumption 2 and Theorem 2 of Ahuja et al. (2020). I did not see any similar theoretical guarantees in this paper. Without them, it is hard to judge whether or not  the proposed method really generalizes out-of-distribution in a sequential manner.
- I do not think the contents in Definition 1&2 are just definitions. Instead, they should be treated as propositions, or lemma or so on, because they should be supported by proofs which seem not to be provided yet.
- The section of Continual IRM by Approximate Bayesian Inference is not friendly with readers unfamiliar with ADMM, which makes that part hard to go through.
- No section index

---

> ### Author Response · Authors · 2020-11-22
> **Thank you for your review - Reply to your Concerns 2, 3 and 4**
>
> We appreciate the positive comments and valuable questions from reviewers. Please let us firstly address your points 2, 3 and 4.
>
> [Q2] I do not think the contents in Definition 1&2 are just definitions. Instead, they should be treated as propositions, or lemma or so on, because they should be supported by proofs which seem not to be provided yet.
>
> [R2] Additional proofs are provided as suggested. In the supplementary material, we added Lemma  6 (Equivalence  of  Definition  1) to highlight the connection of Definition1 and the original IRM definition. For definition 2, this is the use of Eq.7 in Eq.8. We extend the sentence (blue color) after the Definition 2, to justify the definition. We also added Lemma 7(Definition 2) in the supplementary material to reinforce this link. Additionally, we realize that we missed one term in Equation (10a).
> We hope that this additional content addresses your concern.
>
> [Q3]The section of Continual IRM by Approximate Bayesian Inference is not friendly with readers unfamiliar with ADMM, which makes that part hard to go through.
>
> [R3] Yes, there are various details to present and we found it hard to fit the space, we improved the presentation of Equations in section 3.4 and section 3.4.1. We also added a footnote to describe the part of the ADMM in the algorithm 1 and 2.
> If the reviewer feels that we need to further improve the presentation, we could add in the supplementary material a further explanation.
>
> [Q4]No section index
> [R4] We added the numbers to the sections

---

> ### Author Response · Authors · 2020-11-25
> **Thank you for your review - Reply to your Concern 1 on generalization**
>
> Thanks for the valuable suggestions.
>
> We added the generalization analysis in supplementary material A.7, which suggests that the Theorem 9 in Arjovsky et al. (2019) also holds for BIRM and BVIRM.

---

### Official Review · AnonReviewer2 · 2020-10-29
**Interesting paper, needs more polishing**

**Rating:** 6
**Confidence:** 2

**Review:**

## Summary
The paper proposes a generalization of the invariant risk minimization objective to the continual learning setting, where environments are observed sequentially. An ADMM strategy for the solution of the resulting bilevel problem is proposed. In extensive experiments on smaller MNIST-like datasets, the method is shown to perform favorable to recent approaches for continual learning.

## Explanation of Rating
The main strength of the paper is that it attempts to tackle an important and open problem using a reasonably principled approach. The work is well-written, and the methods performs well in practice and is evaluated against a large set of competitors. I lean towards accepting this submission. The weaknesses of the paper are the scalability of the approach (see comment #1) and the lack of theoretical guarantees for the ad hoc ADMM scheme (see comment #2).

## Detailed Comments
1. An evaluation of the method for larger architectures (e.g. a convolutional network) would make the approach more convincing. At the moment, I get the impression that there are issues with the scalability of the approach to larger data sets and models.
2. The ADMM formulation seems rather ad-hoc, and since the loss is not convex, it is unclear whether the scheme is numerically stable / convergent. The statement about convergence rates in the strongly convex / convex setting are a bit puzzling, as they do not apply to the problem at hand. What is the main advantage of the ADMM approach over, say, solving the bilevel problem using implicit differentiation?

---

> ### Author Response · Authors · 2020-11-22
> **Thank you for your review - Reply to your second point**
>
> We appreciate the positive comments and valuable questions from reviewers. Please let us first clarify your second concern.
>
> [Q2] The ADMM formulation seems rather ad-hoc, and since the loss is not convex, it is unclear whether the scheme is numerically stable / convergent. The statement about convergence rates in the strongly convex / convex setting are a bit puzzling, as they do not apply to the problem at hand. What is the main advantage of the ADMM approach over, say, solving the bilevel problem using implicit differentiation?
>
> [R2] Thank you for your comment. Yes, we acknowledge that the use of ADMM is not unique, but it allows us to solve the problem at hand, even if we can not have a general guarantee of its convergence.
> One of the Alternating Direction Method of Multipliers (ADMM) advantages is the ability to scale computationally, as for example in distributed computation. The use of direct bilevel optimization is also a possible direction [1] that we have not explored, which would not be able to scale if computation of environments is done simultaneously. The convergence guarantee is only applicable to local minima, there are no general global convergence results for the full loss function that is not convex, which also applies to SGD based methods. Both IRMv1 and ADMM are implicitly based on the relaxation of the constraint, but the advantage of ADMM is the guarantee of convergence, while in IRMv1 requires heuristics to force the second term to be zero, typically by hand increasing the value of $\lambda$ in Eq.(4) to an arbitrary large value. Finally ADMM naturally solves the BIRM problem and can be extended to the variational case and thus also solving BVIRM.
> We modified the submission in section 3.4 to account with the our comments.
>
> [1] Franceschi, Luca, Paolo Frasconi, Saverio Salzo, Riccardo Grazzi, and Massimilano Pontil. "Bilevel programming for hyperparameter optimization and meta-learning." ICML (2018).

---

> ### Author Response · Authors · 2020-11-25
> **Thank you for your review - Reply to your comment on scalability**
>
> Thank you for your comment, we address here your comment on the scalability.
>
> [Q] An evaluation of the method for larger architectures (e.g. a convolutional network) would make the approach more convincing. At the moment, I get the impression that there are issues with the scalability of the approach to larger data sets and models.
>
> [R] While Variational methods introduce additional cost of constant factor in term of computation cost and parameters space (indeed we are computing a distribution rather then a deterministic network), the bayesian methods are generally applicable to CNN [2] (actually the number of parameters in CNN is typically smaller than in FCN) and their computational complexity does not prevent the use in larger architectures. We have not provided experiments in this direction though. From a practical point of view it is also possible to use a pre-trained (pre-) feature extraction network, followed by the proposed approach. Pre-train network can also be used directly to initialize Variational networks and then followed with the normal training.
>
> [2] Zhao, C., Ni, B., Zhang, J., Zhao, Q., Zhang, W. and Tian, Q., 2019. Variational convolutional neural network pruning. In Proceedings of the IEEE Conference on Computer Vision and Pattern Recognition (pp. 2780-2789).

---

### Author Response · Authors · 2020-11-25
**Summary of the changes**

Dear Reviewers,

We have incorporated your valuable feedback and provided a new paper revision. We summarize here the main changes:

* Updated and corrected definition 2 (section 3.3) and added the connection of Definition 1 and Definition 2 to the original definition of IRM in the Supplementary material (section A.5), supported by associated Lemmas (Lemma 6 and Lemma 7) [per request of R3]
* Clarified the connection of the information projection and updated the description in Section 3.5. In the supplementary material (Section A.6) we provided more insight in the sequential information projection (Section 6.1 and Lemma 8) and its link to discover invariant distributions and IRM (Section A6.2). [per request of R4]
* In section A.7 we extend out-of-distribution generalization results of Arjovsky et al. (2019) (theorem 9) to the BIRM and BVIRM problems. [per request of R3]
* We added the new experimental section 5.3 where we evaluated the Environment Inference for continual learning invariant models, i.e. in presence of a single environment and explored the dependency on the sample size. In the supplementary material we also extended the experiment using the synthetic dataset proposed in Arjovsky et al. (2019) and compared with the method EIIL of  (Creager et al., 2020), using the experimental setup of this last work. [per request of R4]
* We improved the presentation of the ADMM algorithm of section 3.4 and 3.4.1. [per request of R3]

---

### Decision · Program_Chairs · 2021-01-07
**Final Decision**

**Decision:**

Reject

**Comment:**

The authors address the problem of learning environment-invariant representations in the case where environments are observed sequentially.
This is done by using a variational Bayesian and bilevel framework.

The paper is borderline, with two reviewers (R2 and R3) favoring slightly acceptance and two reviewrs (R4 and R1) favoring rejection.

R4 points out that the current experiments do not do a good job of reflecting a continual learning setup and that simple modifications on existing IRM based methods could outperform the method proposed by the authors. The authors are encouraged to take into account the reviewer's
suggestions to improve the paper.

R1 argued initially that the proposed solution is not learning at all since it has errors very close to random guessing. While the authors have improved their method in the revision, the results are still close to random guessing, which questions the practical usefulness of the proposed approach. Also, in the revision, the authors managed to obtain better results when their method is combined with Environment Inference for Invariant Learning (EIIL), but these results are secondary and not the main part of the paper.

The authors should improve the work taking into account the reviewrs' comments.